# Dormancy release of seeds of *Podophyllum hexandrum* Royle accompanied by changes in phytochemicals and inorganic elements

Xijia Jiu[1], Honggang Chen[1,2], Tao Du[1,2]*, XiWei Jia[1], Dong Liu[1], JinJin Meng[1], XiaoJuan Xu[1]

1 College of Pharmacy, Gansu University of Traditional Chinese Medicine, Lanzhou, China, 2 Northwest Chinese and Tibetan Medicine Collaborative Innovation Center, Lanzhou, China

* gslzdt@163.com

## Abstract

*Podophyllum hexandrum* Royle is an alpine medicinal plant of considerable importance, and its seed dormancy severely inhibits population renewal. Although cold stratification can break dormancy to a certain extent, the migration and accumulation of phytochemicals and inorganic elements in the seeds during dormancy release and their functions remain unclear. Changes in phytochemicals and inorganic elements in different seed parts were analyzed during dormancy. The key differential phytochemicals and inorganic elements were screened and their association with dormancy release and their roles in dormancy release were explored. The results showed that dormancy release may have occurred following the decrease in palmitic acid and linoleic acid content in the seeds and the increase in 2,3-dihydro-3,5-dihydro-6-methyl-4 (h)-pyran-4-one content in the endosperm. Meanwhile, 6-propyltridecane and hexadecane in the seed coat may enhance the water permeability of seeds to speed up germination. Mg may migrate from the seed coat to the endosperm and seed embryos, whereas Co may migrate from the seed embryo to the seed coat. Ca, Mn, Mg, and Co are involved in various physiological metabolic processes, which may facilitate the dormancy release of *P. hexandrum* seeds. These findings have enhanced our understanding of the mechanisms of dormancy release in *P. hexandrum* seeds and can serve as a reference for the development of more effective dormancy-breaking techniques for the conservation of this endangered medicinal plant.

**Data Availability Statement:** All relevant data are contained in this manuscript and in the Figshare database, dataset DOI number 10.6084/m9.figshare.24319804.

## Introduction

*P. hexandrum* is a perennial *Podophyllum* L herb that is mainly found in the high Himalayan regions of China, Nepal, Bhutan, northern India, Pakistan, eastern Afghanistan, and Kashmir [1]. Given its outstanding medicinal value, *P. hexandrum* has been used in traditional Chinese and Indian medicine [2–4]. Podophyllotoxin, a lignan component in *P. hexandrum*, is widely used to treat lung cancer, liver cancer, verrucous cancer, and condyloma acuminatum owing to its pharmacological activity. Drugs made from its derivatives, such as etoposide and

**Funding:** This research was Supported by the earmarked fund for CARS-21 and Gansu Science and Technology Innovation Base and Talent Program (18JR2TA017). The funders had no role in study design, data collection and analysis, decision to publish, or preparation of the manuscript.

**Competing interests:** The authors declare that they have no known competing financial interests or personal relationships that could have appeared to influence the work reported in this paper.

teniposide, have been widely used in clinical practice [5–7]. However, the resource reserves of *P. hexandrum* face challenges. This has resulted in its current endangered listing in the Convention on International Trade in Endangered Species of Wild Fauna and Flora (CITES) [8]. It is also listed as a national second-class protected plant in the China Plant Red Data Book [9].

*P. hexandrum* seeds can remain dormant over relatively long timescales. In this context, germination and seedling growth occurs over timescales several months to several years [10]. Population renewal is relatively slow, which can lead to severe resource shortages. Cold stratification is a common method of releasing seed dormancy, which has substantially improved the germination percentage (GP) in *Silene ciliata* and *Linum olympicum* seeds over specific timescales [11, 12]. Dormancy release of *P. hexandrum* seeds can be promoted by cold stratification [13]. From a biomechanical standpoint, seed germination depends on the balance of two opposing forces. One is the force of the seed embryo on the outer top, that is, an increase in the growth potential of the seed embryo. *P. hexandrum* seeds have cryptogeal morphophysiological dormancy [14], with the seed embryo growing to a certain critical length before germination. The second is the reduction in resistance to seed embryo elongation by external coverings such as the endosperm and seed coat, that is, the continued weakening of the endosperm. Proteomic research on seed germination behavior has shown that the hindering effect of the thick-walled endosperm on seed germination is weakened by the increase and accumulation of cell-wall hydrolases [15, 16]. A similar conclusion has been reached in transcriptomic studies, that is, seed germination is facilitated by strongly expressed hydrolase genes [17, 18].

The end of seed dormancy depends on changes in the biochemical levels. Various phytochemicals within the seed, including lipids, starches, proteins, amino acids, and phytohormones, migrate, accumulate, and transform during this process. Amino acids such as arginine serve as nitrogen sources for protein synthesis. Lipids, such as triglycerides, are converted into energy-supporting substances through participation in either the tricarboxylic acid cycle (TCA) or pyruvic acid pathway. Both of these pathways accumulate in large quantities during germination, promoting breaking dormancy [19]. Phytohormones play a crucial role in the regulation of seed dormancy. Gibberellins (GA) and abscisic acid (ABA) are antagonistic. GA promotes seed dormancy by stimulating the synthesis of Aux and cytokinin (CTK) as well as by activating amylase and protease. In contrast, ABA induces dormancy by regulating the accumulation of storage proteins and lipids in the seed [20]. Phytohormones migrate between different tissue parts of the seed. ABA is produced in the endosperm and transported to the seed embryo, whereas GA is produced in the seed embryo and transported to the endosperm during seed germination. GA activates carbon metabolism in the endosperm and facilitates the translation and synthesis of critical proteins for cell growth [21]. The synthesis of phytohormones is usually associated with a number of other metabolites or metabolic pathways. ABA is produced from the key precursor zeaxanthin via a series of epoxidation and isomerisation reactions. GA is produced via a dioxygenase reaction with the key intermediate 2-oxyglutarate in the TCA pathway. In the phytoterpene synthesis pathway, the precursors isopentenyl diphosphate (IPP) and dimethylallyl diphosphate (DMAPP), and the precursors in the plant terpene synthesis pathway, are the basis for the synthesis of these two key phytohormones. Therefore, the synthesis of these two hormones competes with the synthesis of terpenes. Jasmonate (JA) synthesis begins with the formation of 12-oxo-phytodienoic acid from the oxidation of linolenic acid. The synthesis and content of linolenic acid, to a certain extent, determine the level of JA in plants [22]. Many transcriptomic and metabolomic studies have provided evidence for the relationship between phytochemicals and seed dormancy. Studies on *Heracleum moellendorffii* Hance seeds have shown that high expression of enoyl-CoA hydratase promotes the accumulation of fatty acids in the seeds. This facilitates the development of the embryo, and, therefore, accelerates the release of dormancy [23]. Comparative

metabolomic studies of wheat seeds at two levels of dormancy have shown that unsaturated fatty acid analogs, such as cis-vaccenate, oleate, linoleate, and linolenate, undergo accumulation mediated by phospholipase A2 to maintain dormancy. Meanwhile, oxalate regulates the accumulation of fatty acids in seeds, likely through its involvement in ROS metabolism. It is involved in ROS metabolism to regulate seed dormancy, and higher oxalate levels imply an increase in the level of lipid degradation, which promotes the lifting of dormancy [24].

The mechanism by which secondary metabolites regulate dormancy is often unclear, but the current study has provided some direct evidence. The germination of *Sapium sebiferum* seeds is impeded by 2,6-di-tert-butyl-p-cresol, an endogenous component in such seeds [25]. Procyanidins in the seed coat and endosperm cap also play an inhibitory role in seed germination [26]. Research on volatile compounds in *Cuminum cyminum* seeds has highlighted the broad-spectrum growth inhibitory activity of cumin aldehydes in *C. cyminum* on monocotyledons and dicotyledons [27]. Inorganic elements in seeds help to maintain the integrity of the cell membrane and, as part of phytic acid, play an indirect role in the synthesis of proteins and abscisic acid [28]. The accumulation of inorganic elements is also conducive to improving seed vigor [29]. The migration of inorganic elements in seeds creates favorable conditions for endosperm weakening, seed coat rupture, and hypocotyl elongation [30].

Despite the preliminary identification of phytochemicals in *P. hexandrum* seeds, the effects of phytochemicals, accumulation, and migratory release of inorganic elements on *P. hexandrum* seed dormancy release, and how these factors play a role, have not yet been studied. Therefore, in the present study, the dormancy of *P. hexandrum* seeds was released through cold stratification. Changes in the phytochemicals and inorganic elements in different seed parts during dormancy release were then analyzed. Key differential phytochemicals and inorganic elements were screened, their associations with dormancy release were established, and their roles in dormancy release were explored. This study has improved our understanding of the dormancy-release mechanism of the seeds of *P. hexandrum*. Screening for key phytochemicals and inorganic elements can provide a reference for developing more effective dormancy-breaking techniques to conserve this endangered medicinal plant.

## Materials and methods

### Materials

*P. hexandrum* seeds were collected from the He Zheng Botanical Garden (35°15′48″N, 103°24′21″E), Gansu University of Chinese Medicine, Gansu Province, China, in September 2020 and September 2021. They were identified as *P. hexandrum* seeds by Professor Tao Du at the School of Pharmacy, Gansu University of Chinese Medicine. The seeds were soaked in clear water, kneaded to remove pulp, and sieved to remove deflated and defective seeds. The seeds were dried naturally in a cool place and stored in a refrigerator at 4°C for 10 d prior to the next step in the stratification process.

### Instruments and reagents

The instruments used in this study included an iCAP™ RQ inductively coupled plasma mass spectrometer (ICP-MS) system (Thermo Fisher Scientific, USA), an SA-20 atomic fluorescence morphology analyzer (Jitian Instruments Co., Ltd., Beijing, China), an ultra CLAVE microwave digestion system (Milestone, Italy), a VB clave A microwave digestion system (LabTech, Inc., Beijing, China), a DUO-PUR 1308 acid purifier (Milestone, Italy), a LAB 500 CDL glassware washer (Steelco, Italy), a Thermo Scientific™ LabTower™ EDI 30 water purification system (Thermo Fisher Scientific, USA), an 8890A gas chromatography-mass spectrometry (GC-MS) system (Agilent, Agilent, USA), and an AAR S-2 flame atomic absorption

spectrophotometer (Shanghai Electro-Optical Devices Co., Ltd.). Methanol and n-hexane were chromatographically pure, and reagents such as nitric and hydrochloric acids were analytically pure. All the reagents were purchased from Sinopharm Chemical Reagent Co., Ltd.

## Cold stratification of seeds

River sand was washed, sieved (40 meshes), sterilized under high pressure (121˚C, 30 min), and spread in a 3–4 cm layer on the bottom of the container. *P. hexandrum* seeds were then placed in a gauze mesh bag, spread on the river sand in the middle of the container, and covered with another layer of river sand with a humidity of 9–11% at 6–8 cm from the bottom of the container. The container was then sealed with plastic wrap, perforated for ventilation, and placed in a refrigerator at 4˚C. Samples were taken every 15 d and preserved in the refrigerator at −80˚C.

## Extraction and separation of phytochemicials from the seed coat and endosperm

The seed coat and endosperm of the seeds, which were harvested in September 2021, were peeled at each stratification stage with a surgical blade, and 1.0 g of the seed coat and 1.0 g of the endosperm were weighed and placed in a 50 mL centrifuge tube. Next, 25 mL of 80% methanol was added to the tube and ultrasonic extraction (600 W, 40 kHz) was performed for 60 min. After centrifugation at 4000 rpm for 5 min, the supernatant was extracted twice, merged, and equally divided into two portions. One portion was concentrated to dryness by rotary evaporation at 60˚C, redissolved to 5 mL with distilled water, filtered using a 0.22 μm filter membrane, and stored for testing. Other proportions were used to test the biological activity of rapeseed and the information on the samples to be tested is shown in Table 1.

## GC-MS identification of leaching solution

GC-MS was performed on the Agilent 8890A system using an elastic quartz capillary column (HP-5MS, 30 m × 0.250 mm × 0.25 μm) with high-purity He as carrier gas at an injection volume of 1 mL, a flow rate of 1 mL/min, a split ratio of 10:1, and an inlet temperature of 250˚C. The temperature program was described as follows. The initial temperature was 40˚C for 3 min, followed by heating to 42˚C at a rate of 0.3˚C /min for 0 min, then to 125˚C at a rate of 5˚C /min for 0 min, to 90˚C at a rate of 10˚C /min for 0 min, to 175˚C at a rate of 2˚C /min for 10 min, and then to 210˚C at a rate of 1 min for 5 min. The ionization mode was an electronic ignition (EI), electron energy of 70 eV, ion source temperature of 230˚C, quadrupole temperature of 150˚C transmission line temperature of 250˚C, and a scanning mass range of 10–550. The NIST17 spectral library was used for data retrieval and comparison and the NIST Chemistry WebBook database was used for supplementary comparison. The relative retention time

**Table 1. Sample information.**

| Position | Stratification stage | Serial number | Position | Stratification stage | Serial number |
|---|---|---|---|---|---|
| Seed coat | 0 d | Z1 | Endosperm | 0d | R1 |
| | 15 d | Z2 | | 15d | R2 |
| | 30 d | Z3 | | 30d | R3 |
| | 45 d | Z4 | | 45d | R4 |
| | 60 d | Z5 | | 60d | R5 |
| | 75 d | Z6 | | 75d | R6 |
| | 90d | Z7 | | 90d | R7 |

was measured, and the mass spectra were compared for peak confirmation. The relative content of each component, that is, the percentage of the peak area of each component of the total peak area, was determined using the peak area normalization method.

## Determination of element content in the *P. hexandrum* seed coat at different stratification stages

**Preparation of *P. hexandrum* seed coat test solution through microwave digestion.** After the seeds were stratified in 2020, 11 elements (Mg, Ca, V, Mn, Fe, Co, Ni, Cu, Se, Mo, and Zn) in the seed coat at different stratification stages were determined using ICP-MS as follows. 0.1000 g of *P. hexandrum* seed coats at each stratification stage were weighed in a polytetrafluoroethylene digestion tube, which was then added to 3 mL of $HNO_3$ and 1 mL of hydrofluoric acid. The tube cap was screwed down for digestion as per the microwave digestion procedure (Table 2), and the acid was removed at 160°Cm after approximately 1 h, until the remaining liquid in the tube was approximately 1 mL. The solution was transferred to ultrapure water until a constant volume of 50 mL was reached and shaken for assessment. A blank solution was then prepared.

**Preparation of the standard mixed solutions of 11 inorganic elements.** The 11-element mixed standard solution was prepared into the corresponding standard working solution, as follows: 0.1, 0.2, 0.3, 0.4, and 0.5 mL of standard solutions (100 μg/mL) consisting of Mg, Ca, V, Mn, Fe, Co, Ni, Cu, Se, Mo, and Zn were precisely sucked into 100 mL volumetric bottles. The volume was fixed to the scale with ultrapure water. The mixed solutions were shaken well to obtain 100, 200, 300, 400, and 500 ng/mL multi-element standard solutions.

**ICP-MS determination conditions.** The ICP-MS system was operated under the following conditions: radio frequency (RF) power of 1600 W, a cooling gas (argon) flow rate of 13.7 mL/min, an auxiliary gas (helium) flow rate of 0.80 mL/min, and an atomizing gas flow rate of 1.026 mL/min. The linear equations for the standard curves of each element are listed in Table 3. The linear correlation coefficients ($R^2$) of the 11 elements were greater than 0.995, confirming the accuracy and reliability of the results.

## Content determination of five differential elements in different parts of *P. hexandrum* seeds at different stratification stages

Samples from different parts of *P. hexandrum* seeds collected in September 2021 at different stratification stages were weighed to determine the contents of the five different elements. The Ca, Mg, Mn, and Co samples were digested using wetting digestion methods. Their contents were determined using flame atomic absorption spectrometry with reference to GB5009.92–2016, GB5009.241–2017, GB5009.242–2017, and GB/T 13884–2018, respectively. The Se sample was digested using wetting digestion methods, and its content was determined using atomic fluorescence spectrometry, as per GB5009.93–2017.

**Table 2. Microwave digestion procedure.**

| Nr | T (min) | E (W) | T1 (°C) | T2 (°C) | P (bar) |
|----|---------|-------|---------|---------|---------|
| 1 | 10 | 1200 | 140 | 60 | 80.0 |
| 2 | 2 | 1200 | 140 | 60 | 100.0 |
| 3 | 15 | 1200 | 220 | **60** | 130.0 |
| 4 | 2 | 1200 | 220 | 60 | 130.0 |
| 5 | 8 | 1200 | 240 | 60 | 130.0 |
| 6 | 10 | 1200 | 240 | 60 | 130.0 |

**Table 3. Standard curves for the 11 inorganic elements.**

| Inorganic elements | Linear equations | Linear correlation coefficients (R2) |
|---|---|---|
| Mg | Y = 19366.9523X + 253452.7792 | 0.9994 |
| Ca | Y = 2960.5085X + 593424.0436 | 0.9957 |
| V | Y = 49058.3878X + 3512.6615 | 0.9970 |
| Mn | Y = 76771.6969X + 73897.9392 | 0.9979 |
| Fe | Y = 2244.6810X + 120495.7414 | 0.9963 |
| Co | Y = 53257.5106X + 2055.8918 | 0.9970 |
| Ni | Y = 11241.5623X + 2007.3160 | 0.9986 |
| Cu | Y = 26339.7891X + 4721.6267 | 0.9973 |
| Se | Y = 1286.9125X + 2405.2456 | 0.9979 |
| Mo | Y = 14975.3741X + 4349.3479 | 0.9990 |
| Zn | Y = 9886.3293X + 22589.1395 | 0.9996 |

## Determination of water absorption of seeds

A total of 30 *P. hexandrum* seeds collected in September 2021 at each stratification stage were selected and placed in a culture dish filled with distilled water at a constant temperature of 25°C to an extent that the seeds were submerged. The seeds were removed every two hours and weighed immediately after the surface water was absorbed by a piece of absorbent paper until the difference between the two weights was less than 0.0050 g. Water absorption was calculated using the following formula. After fitting using the Boltzmann curve, the water absorption rate is denoted by the center value of the fitted equation. notated as X0.

$$Seed\ water\ absorption(\%) = \frac{Seed\ quality\ after\ water\ absorption - initial\ seed\ quality}{initial\ seed\ quality} \times 100\%$$

## Germination test of *P. hexandrum* seeds

A total of 150 *P. hexandrum* seeds, collected in September 2021, at each stratification stage were randomly selected, disinfected with 1% NaCl for 15 min, washed 3–5 times with distilled water, and sown on a sand bed spread with sand at 3–5 mm. Each dish was sown with 50 seeds in three replicates, cultured in full light at 25°C, and watered (1 mL) daily. The number of germinated seeds was recorded. The GP and time to reach 50% germination (T50) were calculated, where T50 represents the time from the start of seed absorption of water to the time point when the GP reached 50% of the final GP. GP was calculated using the following formula, and T50 was solved after curve fitting.

$$Germination\ percentage(\%) = \frac{Total\ number\ of\ germinated\ seeds}{Number\ of\ seeds\ for\ testing} \times 100\%$$

## Biological activity test of rapeseeds

Leaching solutions (1 mL) of the seed coat and endosperm at different stratification stages were added to culture dishes, where 1 mL of distilled water was added to the control treatment, in three replicates and placed in a fume hood. After complete volatilization of the organic solvent methanol which was dried with a piece of filter paper, 3 mL of distilled water was added into each culture dish. Fifty rapeseeds were randomly selected, spread evenly in the culture dish, and placed in a 3000 Lx light incubator for the germination test at 25°C three times. The number of germinated seeds was recorded daily for three consecutive days until no seeds germinated for three consecutive days. The germination inhibition rate (GIR) and T50 of each

treatment were calculated, and the T50 was determined by linear interpolation. The formula for GIR is as follows, where C denotes the seed GP in the control group, and T stands for the seed GP in each treatment group.

$$GIR = \frac{C - T}{C} \times 100\%$$

## Data analysis

Experimental data were preprocessed using Excel 2016, followed by cluster analysis, analysis of variance (ANOVA), multiple comparisons using SPSS 22, partial least squares discriminant analysis (PLS-DA), and orthogonal partial least squares discriminant analysis (OPLS-DA) using SIMCA 14.1. Correlation analysis was conducted using Chiplot, and the clustering heat map was plotted in OmicShare using Origin 2019.

## Results

### Germination parameters of *P. hexandrum* seeds at different stratification stages

As shown in Fig 1A, the GP of *P. hexandrum* seeds increased from 34.81% to 87.22% with increasing stratification time, and dormancy was gradually released. As illustrated in Fig 1B, the T50 of the seeds at each stage showed an overall downward trend, but the range of

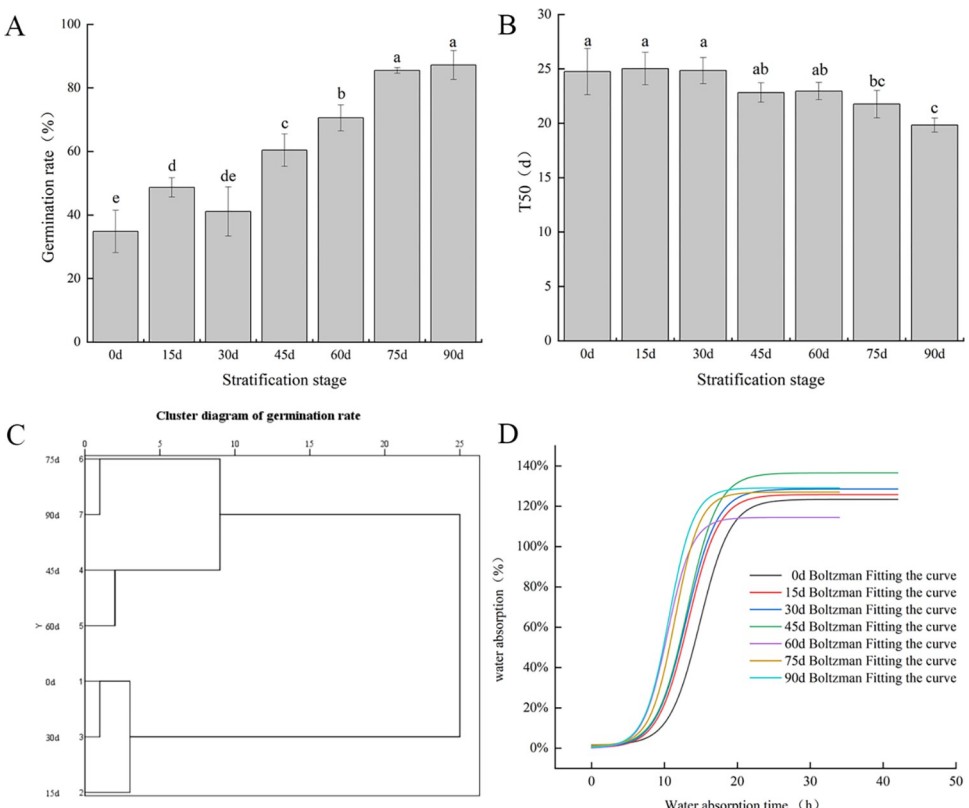

**Fig 1.** (A) Germination rate of *P. hexandrum* seeds at different stratification stages; (B) half germination time of *P. hexandrum* seeds at different stratification stages; (C) clustering results from the seed germination rate at different stratification stages; (D) water absorption curve of *P. hexandrum* seeds at different stratification stages. **Note:** Different lowercase letters indicate significant differences between groups at $p < 0.05$.

**Table 4. Parameters of Boltzman fitting curve equation of seeds in each stage.**

| Stratification stage | A1 | A2 | X0 | dx |
|---|---|---|---|---|
| 0 d | 0.01538 | 1.23422 | 14.8643 | 2.1 |
| 15 d | 5.76344E-4 | 1.2582 | 13.27199 | 2.1 |
| 30 d | 0.00163 | 1.28588 | 13.03232 | 2.1 |
| 45 d | 0.00676 | 1.36579 | 13.12582 | 2.1 |
| 60 d | −0.00127 | 1.14467 | 10.42892 | 1.7 |
| 75 d | 0.0126 | 1.2703 | 11.44779 | 1.7 |
| 90 d | 0.00328 | 1.29139 | 10.61432 | 1.7 |

variation was relatively small. After stratification for 90 d, the T50 of the seeds was significantly lower than that at other stages. The water absorption rates of *P. hexandrum* seeds at different stratification stages were subjected to Boltzmann curve fitting. The cluster analysis results (Fig 1C) showed that the entire stratification process could be divided into three stages, that is, early (0–30 d), middle (45–60 d), and late (75–90 d), according to the degree of dormancy release at a Euclidean distance of 5. The results shown in Fig 1D and Table 4 showed that the water absorption process of seeds at different stages was S-shaped, and the final saturated water absorption rate was similar. However, the water absorption rate of seeds differed at different stages. The central value X0 of the seeds was smaller at 60, 75, and 90 d of stratification, with a faster water absorption rate than that of the seeds at other stages.

## Identification of chemical components in different parts of *P. hexandrum* seeds at different stratification stages

The chromatographic peaks of the 14 seed coat and endosperm samples were matched using the multipoint correction method, and standard fingerprints were automatically generated (Fig 2A and 2B). Twenty and seventeen common peaks were observed in the seed coat and endosperm samples, respectively. The relative retention times of the common peaks were obtained using standard fingerprints, and the spectral libraries of each component were retrieved (Fig 2D, Tables 5 and 6). The endosperm and seed coat were similar to some extent in their standard fingerprints. The common peak components in the seed coat and endosperm were classified and are summarized in Fig 2C. The components of the seed coat and endosperm were found to mainly be alkanes, alkenes, aromatic hydrocarbons, ketones, phenols, alcohols, fatty acids, esters, terpenoids, and steroids, with alkanes (20%), alkenes (10%), phenols (10%), and fatty acids (20%) being the most abundant in the seed coat. However, in the endosperm, alkanes, ketones, fatty acids, and esters played dominant roles, accounting for 82.36%.

## Screening of key differential components in different parts of *P. hexandrum* seeds

**Screening of key differential components in the seed coat.** In this study, a data matrix was established based on the data of the seed coat samples at seven stages. The samples were divided into three stratification stages based on the clustering results under Fig 1C, followed by PLS-DA analysis (Fig 3A). The results have shown that the seed coat samples at the early, middle, and late stratification stages could be distinguished by their compositions. Inter-group differences were observed, with $R^2X = 0.739$ and $R^2Y = 0.929$, both of which were close to 1, and $Q^2 = 0.471 > 0.4$. The differences between samples could be effectively explained and predicted by this model. Model quality was investigated using a 200-time response ranking test

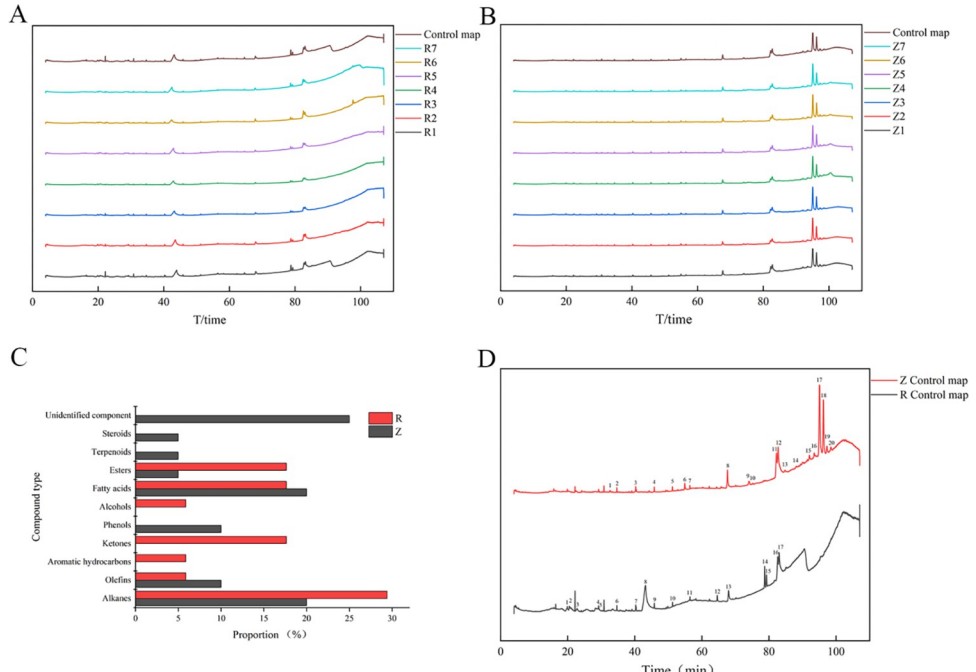

**Fig 2.** (A) Endosperm fingerprint of *P. hexandrum* seed; (B) seed coat fingerprint of *P. hexandrum* seed; (C) classification of compounds from different parts; (D) comparison of standard fingerprints of different parts.

(RPT) to prevent the model from overfitting. As shown in Fig 3B, the results have demonstrated that the $Q^2$ regression line intersects the vertical axis below the zero point. This indicates that the model is accurate and reliable without overfitting. Key components were filtered using the importance (VIP) values of the projection variables in combination with the *p*-value

**Table 5. Endosperm-shared chromatographic peak identification results.**

| Peak number | Retention time | Retrieve RI | Measured RI | CAS number | Components | References |
|---|---|---|---|---|---|---|
| 1 | 19.907 | 1071 | 1096.23 | 13151-34-3 | 3-Methyldecane | |
| 2 | 20.528 | 1110 | 1111.909 | 118-71-8 | 3-Hydroxy-2-methyl-4H-pyran-4-one | |
| 3 | 22.76 | 1151 | 1163.624 | 28564-83-2 | 2, 3-Dihydro-3, 5-dihydroxy-6-methyl-4(H)-pyran-4-one | |
| 4 | 28.27 | 1256 | 1278.346 | 5286-38-4 | 6-methyl-3-(1-methyl ethyl)-7-oxabicyclo[4.1.0]heptan-2-one | |
| 5 | 29.223 | 1300 | 1297.296 | 629-50-5 | Tridecan | |
| 6 | 34.679 | 1400 | 1397.579 | 629-59-4 | Tetradecane | |
| 7 | 40.283 | 1496 | 1497.732 | 55045-10-8 | 6-Propyltridecane | |
| 8 | 43.041 | 1535 | 1547.465 | 4537-11-5 | (1-Butylhexyl)benzene | [31] |
| 9 | 45.844 | 1600 | 1598.034 | 544-76-3 | Hexadecane | |
| 10 | 51.254 | 1702 | 1698.094 | 14852-31-4 | 2-Hexadecanol | |
| 11 | 56.468 | 1793 | 1798.307 | 112-88-9 | 1-Octadecane | [32] |
| 12 | 64.589 | 1984 | 1927.788 | 2490-49-5 | Methyl 14-methylpalmitate | |
| 13 | 67.938 | 1968 | 1969.935 | 57-10-3 | Palmitic acid | |
| 14 | 78.706 | 2097 | 2097.794 | 56554-24-6 | 7,10-Octadecadienoic acid methyl ester | |
| 15 | 79.264 | 2103 | 2104.339 | 1937-63-9 | Cis-11-octadecenoic acid methyl ester | |
| 16 | 82.591 | 2133 | 2143.462 | 60-33-3 | Linoleic acid | |
| 17 | 83.035 | 2162 | 2151.035 | 544-35-4 | Linoleic acid ethyl ester | |

**Table 6. Identification of common chromatographic peaks in the seed coat.**

| Peak number | Retention time | Retrieve RI | Measured RI | CAS number | Components | References |
|---|---|---|---|---|---|---|
| 1 | 32.63 | 1355 | 1359.996 | 91-10-1 | 2,6-Dimethoxyphenol | |
| 2 | 34.669 | 1400 | 1397.395 | 629-59-4 | Tetradecane | |
| 3 | 40.264 | 1496 | 1497.392 | 55045-10-8 | 6-Propyltridecane | |
| 4 | 45.817 | 1600 | 1597.546 | 544-76-3 | Hexadecane | |
| 5 | 51.219 | 1700 | 1697.446 | 629-78-7 | Heptadecane | |
| 6 | 54.924 | 1768 | 1768.609 | 544-63-8 | Myristic acid | |
| 7 | 56.424 | 1795 | 1797.461 | 112-88-9 | 1-Octadecane | [33] |
| 8 | 67.642 | 1968 | 1966.209 | 57-10-3 | Palmitic acid | |
| 9 | 73.99 | 2080 | 2042.752 | 27400-79-9 | henicosene | [34] |
| 10 | 74.174 | 2054 | 2044.9 | 19407-28-4 | Abietatriene | |
| 11 | 82.222 | 2133 | 2139.123 | 60-33-3 | Linoleic acid | |
| 12 | 82.731 | 2141 | 2145.108 | 112-80-1 | Oleic acid | |
| 13 | 84.745 | 2162 | 2168.791 | 544-35-4 | Linoleic acid ethyl ester | |
| 14 | 88.121 | — | 2208.69 | — | — | |
| 15 | 92.046 | — | 2258.341 | — | — | |
| 16 | 93.502 | — | 2273.459 | — | — | |
| 17 | 95.041 | 2280 | 2291.984 | 186099-81-0 | 18-Norandrosta-1,13-dien-3-ol, 17,17-dimethyl-, (3α,5β)- | |
| 18 | 96.201 | 2325 | 2306.096 | 514-62-5 | 8,11,13-Abietatriene-12-ol | [35] |
| 19 | 97.268 | — | 2319.262 | — | — | |
| 20 | 98.467 | — | 2334.057 | — | — | |

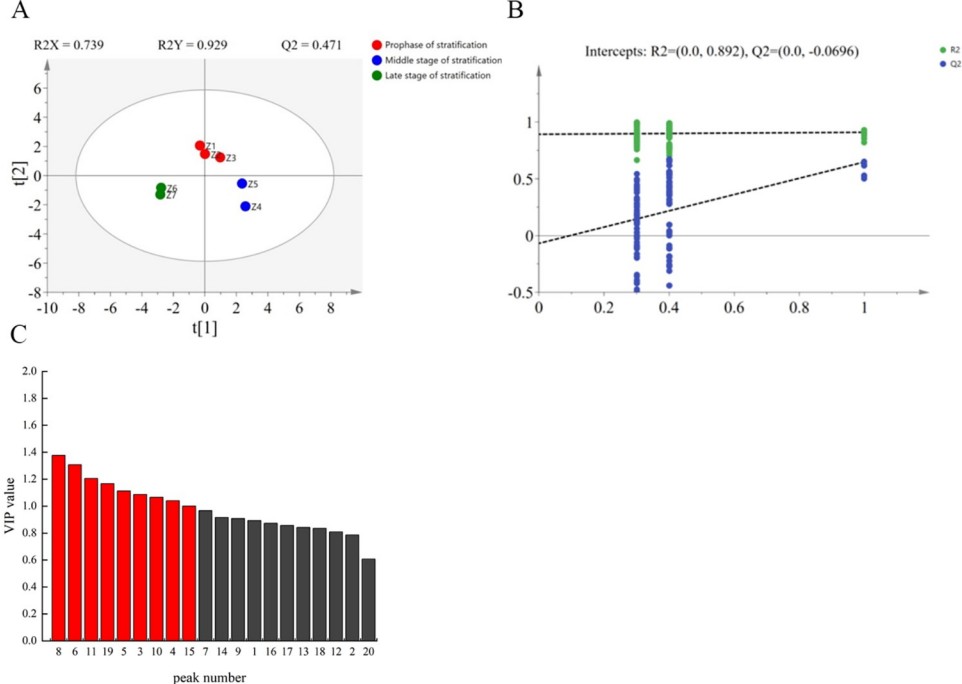

**Fig 3. PLS-DA analysis of seed coat samples of *P. hexandrum* at different stratification stages.** (A) PLS-DA score diagram of seed coat samples at different stratification stages,the circles in the figure indicate the mean values of the samples, larger distances between the circles in the figure indicate larger differences; (B) PRT test diagram of PLS-DA; (C) VIP values of each component.

**Table 7. Key differential compounds of the different parts of the *P. hexandrum* seeds.**

| Position | Peak number | Components | VIP Value | P Value |
|---|---|---|---|---|
| Seed coat | 8 | Palmitic acid | 1.3773 | 0.002 |
| | 11 | Linoleic acid | 1.2055 | 0.002 |
| | 5 | Heptadecane | 1.1124 | 0.008 |
| | 3 | 6-Propyltridecane | 1.0864 | 0.026 |
| | 4 | Hexadecane | 1.0400 | 0.031 |
| Endosperm | 3 | 2, 3-Dihydro-3, 5-dihydroxy-6-methyl-4(H)-pyran-4-one | 1.5944 | 0.015 |
| | 4 | 6-methyl-3-(1-methylethyl)-7-oxabicyclo[4.1.0]heptan-2-one | 1.3158 | 0.023 |
| | 16 | Linoleic acid | 1.2502 | 0.002 |
| | 13 | Palmitic acid | 1.0178 | 0.028 |
| | 15 | Cis-11-octadecenoic acid methyl este | 1.0174 | 0.571 |

with the VIP value >1 and *p*-value < 0.05. As shown in Fig 3C and Table 7, palmitic acid, linoleic acid, heptadecane, 6-propyl tridecane, and hexadecane were the key components used to distinguish the three stratification stages.

**Screening of key differential components in the endosperm.** Given the poor ability of PLS-DA to distinguish between the three stratification stages of the endosperm, the samples were further differentiated using the OPLS-DA method. As a multivariate statistical analysis method with supervised mode recognition, OPLS-DA can filter noise unrelated to classification information to the maximum extent, magnify the difference between samples, and optimize the relationship model between sample groups. As shown in Fig 4A, the samples at the three stratification stages were differentiated using OPLS-DA, with $R^2 = 0.839$ and $Q^2 = 0.55$, indicating that the model was robust and reliable. The 200-time RPT results (Fig 4B) illustrated that the $Q^2$ regression line intersected with the Y-axis on the negative half axis. This indicates

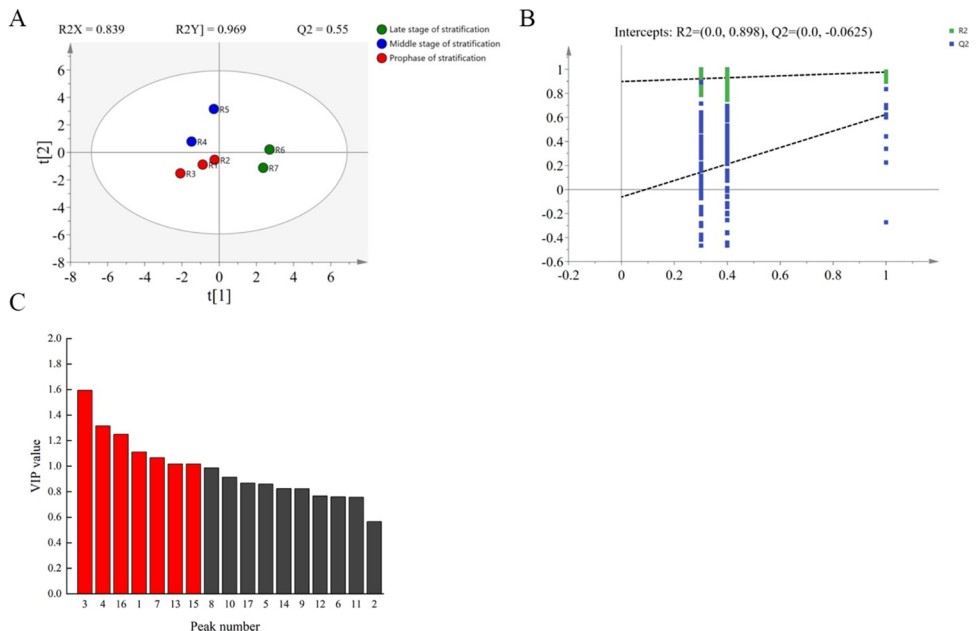

**Fig 4. OPLS-DA analysis of endosperm samples of *P. hexandrum* at different stratification stages.** (A) OPLS-DA score diagram of Endosperm samples at different stratification stages, the circles in the figure indicate the mean values of the samples, larger distances between the circles in the figure indicate larger differences;(B) PRT test diagram of OPLS-DA; (C) VIP values of each component.

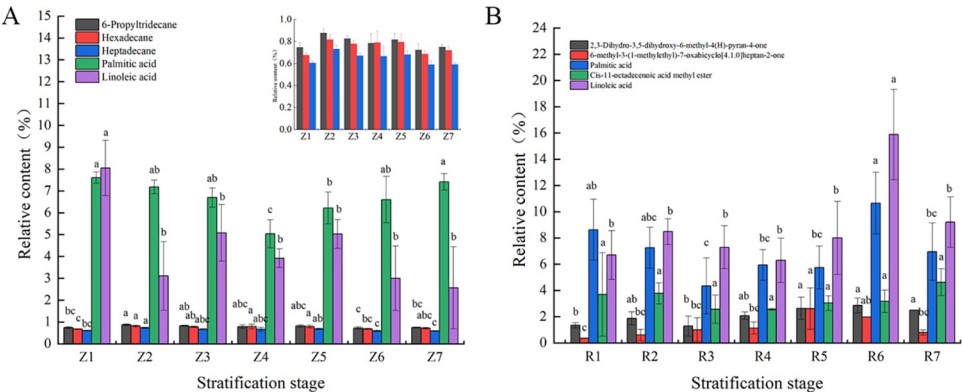

**Fig 5. Changes in the content of key differential components of *P. hexandrum* seeds at different stages of stratification.** (A) Changes in the contents of key differential components in the seed coat; (B) changes in the contents of key differential components in the endosperm. **Note:** Different lowercase letters indicate significant differences between groups at $p < 0.05$.

that variable information was fitted using the established OPLS-DA model, which performed well in prediction. The VIP method and $p$-value were combined for screening, and five key differential compounds were obtained (Table 7), namely, 3,5-dihydroxy-6-methyl-2H-pyran-4 (3H)-one, 6-methyl-3-(1-methyl ethyl)-7-oxabicyclo [4.1.0] heptane-2-one, linoleic acid, palmitic acid, and cis-11-octadecenoic acid methyl ester.

## Changes in the content of key differential components in different parts of *P. hexandrum* seeds

The changes in the 10 differential components in the seed coat and endosperm during the entire stratification process (Fig 5) were compared, and the results showed that linoleic acid and palmitic acid played dominant roles in the seed coat and endosperm. Throughout the stratification process, the linoleic acid content of the seed coat decreased significantly, from 8.05% to 2.56%. However, un the endosperm, no significant differences in the linoleic acid content were observed, except that it increased significantly at 75 d. The linoleic acid content in the endosperm was substantially higher than that in the seed coat, with a minimum content of 6.7%. This approached the highest linoleic acid content (8.05%) in the seed coat. During stratification, the palmitic acid content in the seed coat was similar to that in the endosperm, and they showed similar variation trends; that is, both first declined and then increased, with a small variation range. The relative contents of the other three key differential components— 6-propyl tridecane, hexadecane, and heptadecane—in the seed coat were low, all being 0.4– 1%, showing an initial increasing and then declining trend and peaking at 45–60 d of stratification. The other three key components—3,5-dihydroxy-6-methyl -2H- pyran -4(3H)-one, 6-methyl -3-(1-methyl ethyl)-7-oxabicyclo [4.1.0] heptane-2-one, and cis-11-octadecenoic acid methyl ester—in the endosperm had a relative content of 0.5–2%. Among these, the content of the two components showed a rising trend with the stratification time and peaked at 60–90 d, except for there being no significant changes in the cis-11-octadecenoic acid methyl ester content.

## Effects of the leaching solution on the germination of rapeseeds

The effects of seed-leaching solutions from different parts of *P. hexandrum* at different stages of rapeseed germination were determined to further explore the influence of phytochemicials in *P. hexandrum* seeds on seed germination. As shown in Fig 6, the GIRs of the endosperm

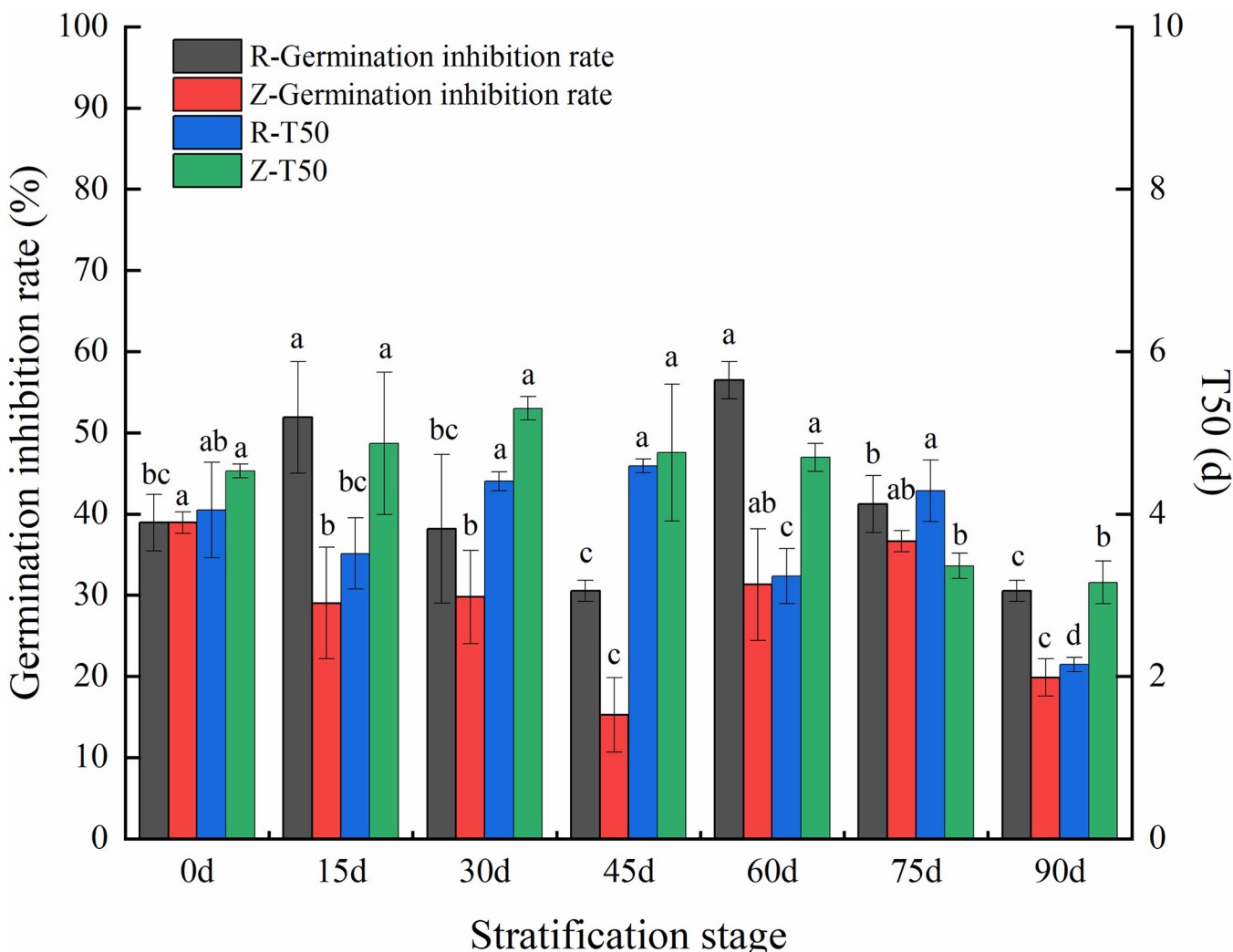

**Fig 6. Changes in inhibition rate and half germination time of rapeseeds. Note:** Different lowercase letters indicate significant differences between groups at $p < 0.05$.

and seed coat leaching solutions for rapeseeds decreased first, then increased, and decreased again with stratification time, reaching their minimum values at 45 and 90 d of stratification. This indicates that the leaching solutions exerted the weakest inhibitory effects on rapeseed germination at the two stages. At different stratification stages, the endosperm leaching solution exerted relatively consistent influences on the T50 of rapeseeds. This however, declined significantly at 60 and 90 d ($p<0.05$) and already dropped to 2.15 d at 90 d of stratification, approaching that (2.26 d) of CK. The T50 of the seed coat leaching solution in the first five stages showed no significant differences ($p>0.05$), but decreased in the last two stratification stages ($p<0.05$).

### Correlation analysis between key differential components and germination parameters of *P. hexandrum* seeds and rapeseeds

Correlations of seed GP and T50 with 10 key differential phytochemicals in different plant parts were analyzed. As shown in Fig 7A, the correlation coefficient of seed GP with Z-(linolic

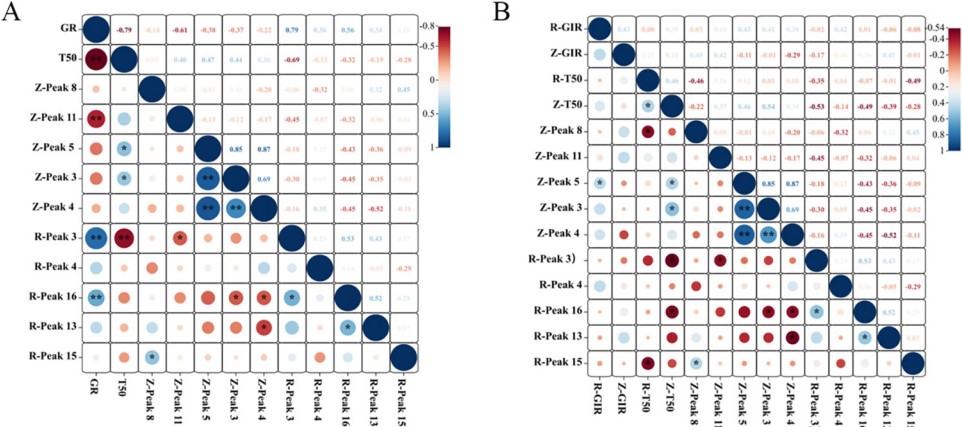

**Fig 7.** (A) Correlation analysis between key differential components and seed germination parameters of *P. hexandrum*; (B) correlation analysis between key differential components and *Brassica rapa* seed germination parameters. **Note:**The scales on the right side of the figure indicate the color shades corresponding to different correlation coefficients,the size of the circle indicates the size of the *p*-value.

acid) was −0.612, indicating a highly significant negative correlation, whereas that with r-(2,3-dihydro-3,5-dihydro-6-methyl-4 (h)-pyran-4-one) was 0.791, indicating a highly significant positive correlation. The correlation coefficient with R-(linolic acid) was 0.560, indicating a significant positive correlation. T50 was significantly correlated with Z-(6-Propyltridecane) and Z-(hexadecane), except for R-(2,3-dihydro-3,5-dihydro-6-methyl-4 (h)-pyran-4-one). Therefore, the release of seed dormancy and germination may be promoted and accelerated by the accumulation of 2,3-dihydro-3,5-dihydro-6-methyl-4 (h)-pyran-4-one. However, the germination process may be hindered by the accumulation of 6-propyltridecane and hexadecane. Rapeseeds were subjected to further biological activity tests, and the correlations of GIR and T50 of rapeseeds extracted from the endosperm and seed coat with ten key phytochemicals in different parts were analyzed. The results showed a significant positive correlation between R-(GIR) and Z-(heptadecane) with a correlation coefficient of 0.43 (Fig 7B). There were no significant correlations between Z-(GIR) and any of the items. R-(T50) was significantly negatively correlated with Z-(palmitic acid) and R-(cis-11-octadecadienoic acid methyl ester), [correlation coefficients: −0.462 and −0.486]. Z-(T50) was significantly correlated with Z-(heptadecane), Z-(6-propyltridecane), R-(2, 3-dihydro-3, 5-dihydroxy-6-methyl-4(H)-Pyran-4-one), and R-(linoleic acid). Z-(T50) presented significant correlations with four items, namely, Z-(heptadecane), Z-(6-propyltridecane), R-(2, 3-dihydro-3, 5-dihydroxy-6-methyl-4(H)-pyran-4-one), and R-(Linoleic acid). Here, Z-(T50) showed significantly positive correlations with Z-(heptadecane) (*p* = 0.46) and Z-(6-propyltridecane) (*p* = 0.54) and significantly negative correlations with R-(2, 3-dihydro-3, 5-dihydroxy-6-methyl-4(H)-pyran-4-one), and R-(linoleic acid). However, only numerical correlations were found between R-(GIR) and Z-(heptadecane), R-(T50) and Z-(palmitic acid), and Z-(T50), R-(2, 3-dihydro-3, 5-dihydroxy-6-methyl-4(H)-pyran-4-one), and R-(linoleic acid), without practical significance. This has further demonstrated that seed germination may be hindered by the accumulation of 6-propyltridecane. Accumulation of cis-11-octadecenoic acid methyl ester may accelerate seed germination.

## Inorganic element content in the seed coat of *P. hexandrum* at different stratification stages

Table 8 shows that the content of 11 inorganic elements had a certain variation trend as the stratification stage progressed. Among these, Ca and Mg reached the highest content in the

**Table 8. Contents of seven intradermal inorganic elements in *P. hexandrum* seed at different stratification stages.**

| Element (µg/g) | Stratification stage | | | | | | |
|---|---|---|---|---|---|---|---|
| | 0 d | 15 d | 30 d | 45 d | 60 d | 75 d | 90 d |
| Mg | 773.40 | 813.06 | 819.61 | 771.59 | 1046.55 | 746.83 | 952.94 |
| Ca | 1188.04 | 2108.03 | 2155.35 | 2082.12 | 2591.32 | 2143.08 | 2815.07 |
| V | 0.5 | 0.76 | 0.81 | 1.38 | 2.41 | 0.67 | 1.16 |
| M | 18.49 | 17.61 | 18.82 | 16.66 | 26.68 | 16.94 | 25.74 |
| Fe | 266.16 | 335.49 | 330.64 | 658.3 | 544.37 | 275.93 | 350.97 |
| Co | 0.21 | 0.24 | 0.36 | 0.43 | 0.65 | 0.24 | 0.56 |
| Ni | 2.87 | 1.62 | 1.64 | 1.87 | 2.64 | 1.31 | 2.35 |
| Cu | 13.81 | 11.53 | 8.34 | 9.11 | 12.57 | 9.44 | 11.02 |
| Zn | 82.76 | 58.11 | 86.69 | 104.66 | 82.8 | 98.5 | 115.28 |
| Se | 0.04 | 0.09 | 0.08 | 0.03 | 0.02 | 0.02 | 0.03 |
| Mo | 1.18 | 0.87 | 0.83 | 1.1 | 0.77 | 0.74 | 1.08 |

seed coat, presenting an upward trend with the lengthening of stratification time. Their contents after stratification were 1.23-fold and 2.37-fold, respectively. The contents of trace elements such as Fe, Mn, Zn, and Cu in the seed coat were relatively high. The contents of Fe, Mn, and Zn generally showed a rising–falling–rising trend throughout the stratification process. Meanwhile, the Cu content fluctuated to some extent, but with a general steady variation trend. The other beneficial trace elements were less abundant in the seed coat, except that the contents of V and Co increased with stratification time, whereas those of Ni, Se, and Mo fluctuated slightly, showing an overall steady change.

## Principal component analysis (PCA) and partial least squares discriminant analysis (PLS-DA) of 11 inorganic elements

PCA is an unsupervised data analysis method. In the PCA diagram, the distance between different samples can be observed through a scatter diagram of the sample distribution. The similarities between different samples can be reflected by the distance between sample groups. The greater the distance, the larger the difference, which can be used to evaluate the stability of the detection system. PCA was conducted for samples at different stratification stages to differentiate between the inorganic elements at different stratification stages. As shown in Fig 8A, the samples in the same group were close to each other on the score map, with similarities. However, some sample groups were not significantly different because of the small intergroup distance. Therefore, they could not be distinguished clearly. Dormancy of *P. hexandrum* seeds starts to be released after 60 d of stratification [13]. Therefore, the entire stratification stage of the samples was divided into dormancy and dormancy release stages, with 60 d as the threshold. Therefore, PLS-DA was performed. PLS-DA, a supervised discriminant analysis method, can filter noise unrelated to classification information, further optimize the relationship model between sample groups, highlight their differences, and compensate for the deficiencies of the PCA method. As shown in Fig 8B, the samples were well-distinguished by PLS-DA, with $R^2$ and $Q^2$ values of 0.924 and 0.86, respectively. Both of these were close to 1, indicating that the differences between the two sample groups could be more effectively explained and predicted by this model. To prevent model overfitting, the model quality was examined using the 200-time RPT method. The results are shown in Fig 8C. All the $Q^2$ values on the left side were lower than the original $Q^2$ values on the right side. The $Q^2$ regression line intersected with the vertical axis below the zero point, indicating that the model was not overfitted and that it was established accurately and reliably.

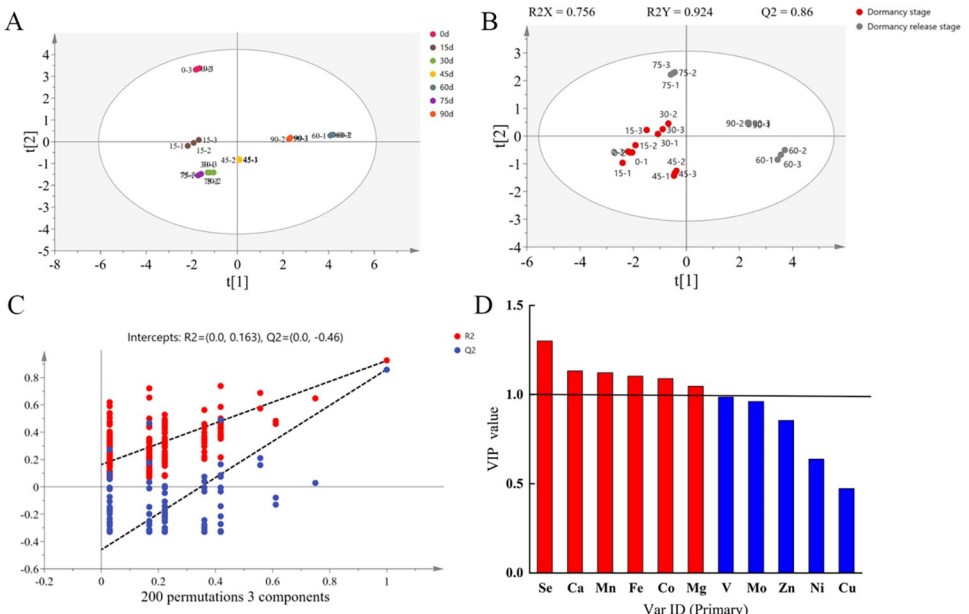

**Fig 8. Multivariate statistical analysis of 11 inorganic elements in *P. hexandrum* seed samples at different stages of stratification.** (A) PCA scores of samples at different stratification stages, circles of the same color in the figure indicate three replicates in the sample, larger distances between the circles in the figure indicate larger differences; (B) PLS-DA scores between samples before and after dormancy release; (C) PRT test diagram of PLS-DA; (D) VIP values of inorganic elements.

According to the PLS-DA, the VIP values of the projection variables could be used to reflect their degree of contribution. A VIP value greater than 1 indicates that the variable is important, with a high degree of contribution. As shown in Fig 8D, six key differential inorganic elements, namely, Se, Ca, Mn, Fe, Co, and Mg, were selected according to the VIP values. Meanwhile, the two groups of samples were tested using the t-test, and the differential inorganic elements were screened according to the *p*-value with both $p < 0.05$ and VIP > 1 as the criteria. As shown in Table 9, five key differential inorganic elements were selected by combining two methods, that is, two macroelements (Ca and Mg) and three trace elements (Mn, Se, and Co).

## Clustering and correlation analysis between key differential elements

Based on the clustering basis of the screened key differential inorganic elements, seed coat samples of *P. hexandrum* at different stratification stages were subjected to cluster analysis via OmicShare from two dimensions, that is, the inorganic elements and stratification stages. As shown in Fig 9A, the abscissa represents different stratification stages and the ordinate

**Table 9. Key differential inorganic elements in seed coat before and after dormancy release of *P. hexandrum* seed.**

| Key Differential Inorganic Elements | VIP Value | P Value |
|---|---|---|
| Mn | 1.1229 | 0.010 |
| Ca | 1.1332 | 0.001 |
| Mg | 1.0468 | 0.026 |
| Co | 1.0897 | 0.029 |
| Se | 1.3004 | 0.001 |

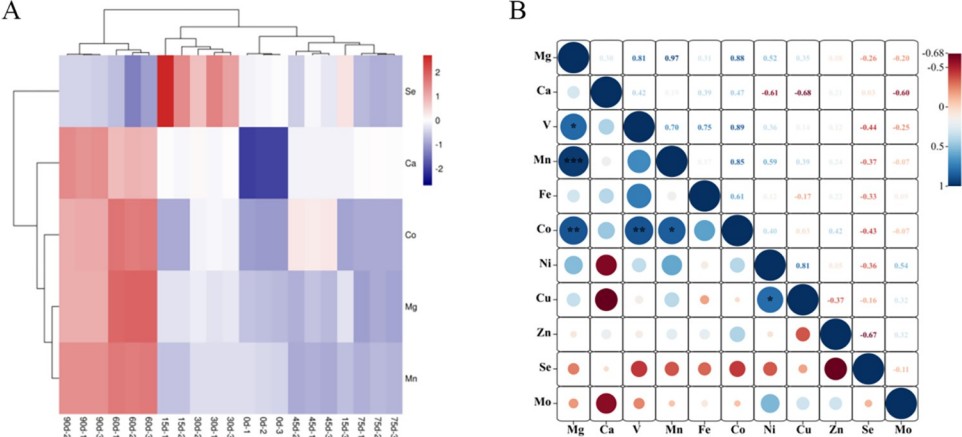

**Fig 9.** (A) Cluster analysis of differential inorganic elements, shades of color in the figure indicate differences in content; (B) correlation analysis of differential inorganic elements, the scales on the right side of the figure indicate the color shades corresponding to different correlation coefficients, the size of the circle indicates the size of the *p*-value.

represents five differential inorganic elements. The red color denotes a significant increase in inorganic elements and the blue color indicates a significant decrease in inorganic elements. As shown in Fig 9A, the two stratification stages, namely 60 d and 90 d, were clustered together. The other stages were clustered together, indicating that the samples before and after dormancy release could be distinguished by the selected differential inorganic elements. Among these elements, Ca, Co, Mg, and Mn accumulated during the dormancy release stage (60 and 90 d), whereas Se accumulated during the dormancy stage (15 and 30 d) and then showed a downward trend. This indicates that different inorganic elements responded significantly differently to dormancy release. The correlation analysis results (Fig 9B) showed that five groups of the 11 inorganic elements presented significantly positive correlations with five different inorganic elements, namely Mg–Co, Mg-Mn, Mg-V, Mn-Co, and Co-V, indicating that their cumulative migration in seeds may have a synergistic effect.

## Changes in the content of five key differential elements in different parts of *P. hexandrum* seeds

To further investigate the distribution and variation of inorganic elements in *P. hexandrum* seeds during stratification, the contents of five key differential elements in different parts, namely, the seed coat, endosperm, and embryo, were measured. The results are presented in Fig 10. Se was not detected in all parts at any stage, and Mg and Ca were mainly enriched in the seed coat, but less in the embryo and endosperm. The Mg content of the seed coat decreased during the first two stratification stages ($p<0.05$). At a later stage, the Mg content in the embryo first decreased, then increased, reaching a minimum at 45 d of stratification. The Mg content in the endosperm increased with stratification time and accumulated significantly at 45 d and 90 d of stratification ($p<0.05$). The Ca content in the embryo declined significantly ($p<0.05$) from 45 to 75 d Meanwhile, the Ca content in the seed coat changed relatively little from 2243.92 to 2010.67 µg/g after stratification. The Ca content in the endosperm showed an overall downward trend, with the lowest content at 45 and 90 d. Mn and Co reached their highest content in the embryo, followed by that of the seed coat and endosperm. The Mn content in the embryo decreased at first and then increased, reaching the lowest content of 51.5 µg/g at 45 d of stratification. However, no significant difference was observed before and after stratification ($p>0.05$). The Mn content in the seed coat declined significantly after

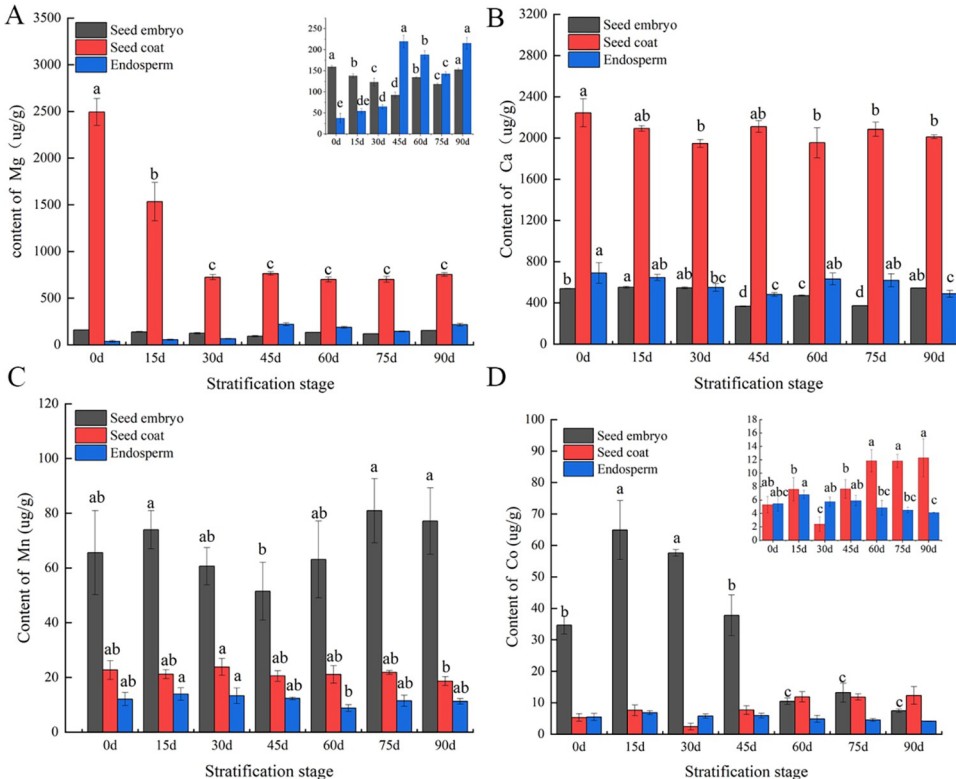

**Fig 10. Changes in the contents of key differential elements in *P. hexandrum* seeds at different stages of stratification.** (A) Mg element content change; (B) Ca element content change; (C) Mn element content change; (D) Co element content change. Note: Different lowercase letters indicate significant differences between groups at p< 0.05.

stratification for 90 d ($p<0.05$), and no significant differences were observed among the other stages. Co significantly accumulated in the embryo at 15 d of stratification, then decreased gradually, and dropped to 7.43 µg/g at the 90 d. Meanwhile, its content in the seed coat increased, and it accumulated at the late stratification stages (60, 75, and 90 d) ($p<0.05$). The Co content in the endosperm changed slightly and decreased to 4.15 µg/g at 90 d of stratification.

## Correlation analysis of differential elements in different parts of *P. hexandrum* seeds with their germination parameters and water absorption characteristics

The correlations of GP, T50, water absorption, and water absorption rate with Ca, Mg, Mn, and Co in different seed parts were explored through correlation analysis. As shown in Fig 11, seed GP was positively correlated with Co in the seed coat and Mg in the endosperm, but negatively correlated with Co in the embryo. T50 was positively correlated with Co in the embryo and endosperm and with Mn in the seed coat, but it was negatively correlated with Co and Mg in the seed coat. No significant correlations were found between the water absorption rate of the seeds and elements in the different plant parts. The water absorption rate (X0) of seeds was found to be significantly correlated with Co and Mg in the seed coat. Among these, its correlation coefficient with Co in the seed coat was −0.81, indicating a significantly negative correlation, and there was a significantly positive correlation with Mg in the seed coat. These results

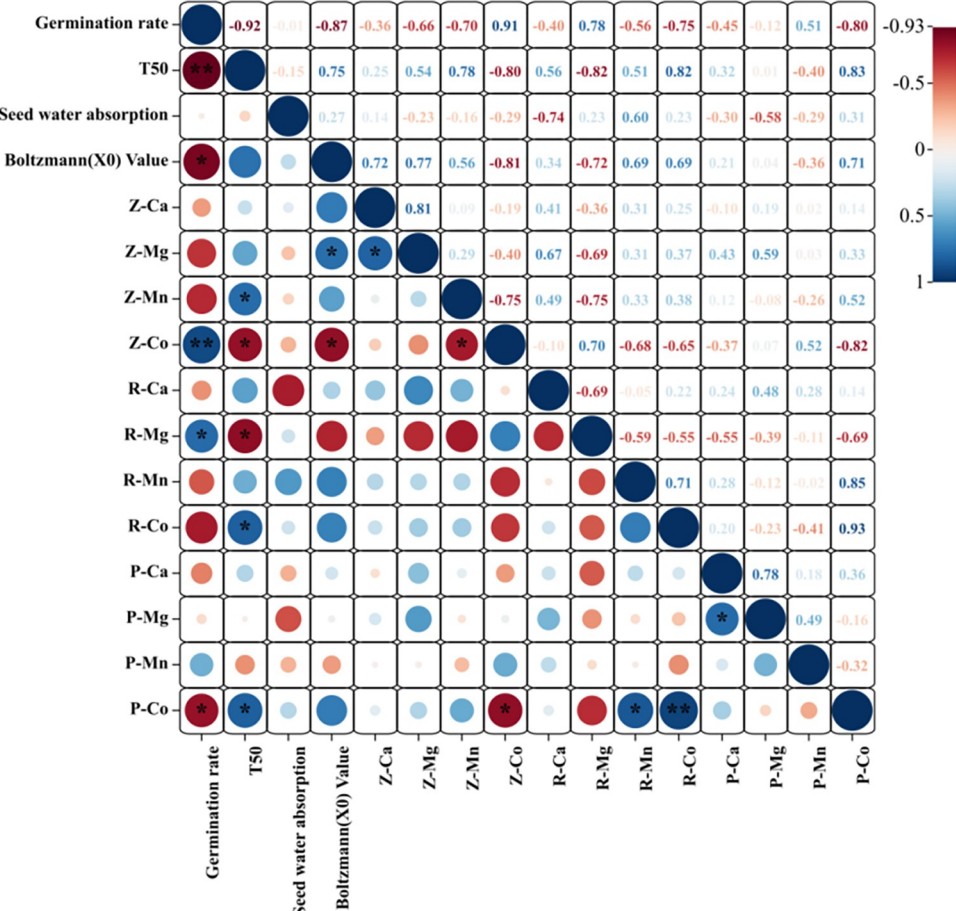

**Fig 11. Correlation analysis between key difference elements and germination parameters and water absorption parameters. Note:** The scales on the right side of the figure indicate the color shades corresponding to different correlation coefficients, the size of the circle indicates the size of the P-value.

have shown that if accumulated, Mg in the seed coat and endosperm may accelerate water absorption in seeds and play a positive role in dormancy release. An opposite correlation was observed between Co in different plant parts and seed GP, indicating that dormancy release is affected differently by different Co content ranges.

## Discussion

Low-temperature stratification is a well-established and efficient method for breaking seed dormancy, particularly physiological dormancy. We assessed the dormancy levels of seeds at various stratification durations using four indices, that is, the germination rate, germination time lag, water absorption rate, and water absorption rate. We discovered that 90 d of low-temperature stratification effectively lifted the dormancy of *P. hexandrum* seeds. The entire stratification process could be divided into three levels based on the degree of dormancy. The seed germination rate could reach 87.22% after 90 d of stratification, and with the gradual lifting of dormancy, the time lag of germination was shortened and the rate of water uptake was accelerated. This indicated that the permeability of the seed coat increased, and the improved germination ability was also an indicator of an increase in the level of physiological metabolism within the seeds.

Phytochemicals in plants affect seed germination; terpenes, ketones, phenols, and fatty acids are common examples. Previous studies have provided direct evidence of this. Eight sesquiterpenes in the *Acorus calamus* seeds inhibit the germination of lettuce seeds [36]. The ketones (1S, 6R)-2, 7 (14), and 10-bisaboltrien-1-ol-4-one in *Cryptomeria japonica* seeds display inhibitory activity against lettuce and rice seeds [37]. α-pyrone derivatives can completely inhibit the germination of *Amaranthus caudatus* seeds [38]. Catechin, a phenolic component in *Lespedeza* seeds, represses seed germination [39]. Seed dormancy is also influenced by phytochemicals in the seeds. Dormancy of lettuce seeds may be induced by the accumulation of coumarin [40]. Meanwhile, dormancy of *Arabidopsis thaliana* seeds is promoted by 12-oxo-plant dienoic acid and abscisic acid [41]. The composition of phytochemicals in the seeds varies considerably among different plants. Phenols and esters are mainly contained in the seed coat of *Triticum aestivum* [42]. Meanwhile, phenolic acids, such as p-hydroxybenzoic acid and protocatechuic acid, play a dominant role in the seed coat of *Parthenium argentatum* [43]. According to the GC-MS analysis of methanol extracts from the seed coat and endosperm at different stratification stages, the phytochemicals were mainly alkanes and fatty acids, the types and contents of which varied with different seed parts. More phenols were detected in the seed coat, whereas more ketones and esters were present in the endosperm. This may have had different effects on dormancy release. Following multivariate statistical analysis, five key differential components were filtered from the seed coat and endosperm and were found to be ketones, esters, alkanes, and fatty acids. A variety of derivatives with pyranone as the parent nucleus exhibit strong free radical scavenging and antioxidant functions [44].Separation of fractions and evaluation of antioxidant activity of leaf extracts of *Odontonema strictum* showed that 6-substituted 5,6-dihydro-α-pyrones were the major components exerting antioxidant activity [45]. The activities of superoxide dismutase, peroxidase, and catalase were significantly higher in the exogenous pyrantel treatment group than the control [46]. Reactive oxygen metabolism play an important role in the regulation of seed dormancy. The biosynthesis of GA is stimulated by ROS through mitogen-activated protein kinase (MAPK) cascades. The accumulation of $H_2O_2$ causes ABA degradation through influencing ABA catalytic enzyme [47]. Therefore it is important to maintain the homeostasis of reactive oxygen species in seeds. In this study, we found that the ketone components 2,3-dihydro-3,5-dihydroxy-6-methyl-4 (H)-pyran-4-one and 6-methyl-3-(1-methylethyl)-7-oxabicyclo[4.1.0]heptan-2-one in the endosperm of *P. hexandrum* seed showed an increasing trend at the late stage of stratification. Oxabicyclo[4.1.0]heptan-2-one both showed an increasing trend in the late stage of stratification. This trend was significantly positively correlated with their seed germination rates and significantly negatively correlated with T50. From this, we deduced that the seeds of *P. hexandrum* showed an increasing trend in the late stage of stratification through the accumulation of ketone constituents, 2,3-dihydro-3,5-dihydroxy-6-methyl-4(H)-pyran-4-one and 6-methyl-4 (H)-pyran-4-one. (H)-pyran-4-one and 6-methyl-3-(1-methylethyl)-7-oxabicyclo[4.1.0]heptan-2-one, therefore, enhancing their own antioxidant capacity, eliminating excessive reactive oxygen species during metabolism, maintaining reactive oxygen species homeostasis, and accelerating the rate of seed germination and facilitating the lifting of dormancy. Recent experimental data (unpublished) also showed that SOD activity and CAT activity within the endosperm of *P. hexandrum* seeds would rise significantly in the late stage of stratification, which supports this inference. Alkanes are the main components of cuticle wax [48, 49]. As a hydrophobic layer, wax usually prevents water uptake, which affects germination. In this study, we found that the contents of alkane components 6-propyltridecane and hexadecane in the seed coat of *P. hexandrum* showed a decreasing trend during stratification and were positively correlated with the T50 of *P. hexandrum* seeds. We hypothesized that the decrease in the contents of 6-propyltridecane and hexadecane resulted in the loosening of the cuticle waxes and

enhanced the hydrophilicity of the seed coat. This, in turn, increased seed permeability and promoted germination. Alkane synthesis is regulated by phytohormones and transcription factors, and ABA and Melatonin act synergistically to promote alkane synthesis in watermelon leaves under drought stress [50]. The mechanism involved may be ABA promoting alkane biosynthesis by negatively regulating the transcriptional repressors of BnCER1-2, BnaC9. DEWAX1. This, in turn, promotes alkane biosynthesis [51]. Changes in phytohormones within the seeds of *P. hexandrum* during low-temperature stratification [13] had an overall decreasing trend in the ABA content. This is also in line with the results that we have obtained thus far. However, further experiments are needed to verify whether ABA mediates the synthesis of alkane constituents, such as 6-propyltridecane and hexadecane. This, in turn, modulates the translucency of the seed coat of *P.hexandrum*. Fatty acids are thought to act as chemosensitizers that affect seed germination. Short-chain fatty acids inhibit the germination of *Cucumis sativus* seeds [52] and maintain the dormancy of *Avena fatua* seeds [53]. Meanwhile, the premature germination of *Mangifera indica* seeds results from a reduction in long-chain fatty acid synthesis [54]. The exogenous application of palmitic and linoleic acid monomers significantly inhibited the germination of *Lolium multiflorum* and *Phalaris minor* seeds, and the inhibitory ability increased with increasing concentration [55]. In this study, the inhibitory effects of the seed coat extract and endosperm extract of *P.hexandrum* seed on the germination of oilseed rape seeds at different stages of stratification were consistent with the changes in contents of palmitic acid and linoleic acid in their corresponding parts. The contents of palmitic acid and linoleic acid accounted for a larger proportion in both the seed coat and the endosperm. Therefore, it is likely that these two phytochemicals are the main germination inhibitors in *P. hexandrum* seed. Saccharides are the end products of fatty acid decomposition. Through a series of pathways such as oxidation, fatty acids, the glyoxylate cycle, and the tricarboxylic acid cycle, ultimately generate glucose and sucrose by gluconeogenesis. This can be used as a carbon source and an energy donor to directly promote seed germination of seeds [56]. We hypothesized that the reduced linoleic and palmitic acid content in the seed coat and endosperm of *P. hexandrum* seeds is broken down into soluble sugars for direct energy supply. This leads to a significant increase in seed germination in the later stages of stratification.

The release of seed dormancy is concurrent with intense physiological activities and energy metabolism, such as the decomposition of substances, such as starch, lipids, and proteins, during storage, the activation of related metabolic enzymes, and the synthesis of physiologically active substances [57], with which the physiological functions of inorganic elements are often closely associated. We first determined the changes in the contents of 11 elements, including Mg, Ca, Mn, Zn, and Cu, which are commonly thought to be beneficial elements, in *P. hexandrum* seed coats at various stages of stratification. We then screened out the five differential inorganic elements, that is, Ca, Mg, Mn, Co, and Se, by combining the PLS-DA analysis with the T-test. This suggested that these five elements may play an important role in the process of dormancy lifting. Correlation research also showed that there may be a synergistic effect between Mg, Mn, and Co. However, it is still unclear how inorganic elements move and accumulate in different sections of the seed. Therefore, we determined the content of the five main differential elements in stages and parts. The Mg element in seeds was primarily enriched in the seed coat prior to stratification. However, the Mg content in the seed embryo and endosperm increased significantly in the late stage of stratification. This suggests that there was a trajectory of Mg migration from the outer seed coat to the inner seed embryo and endosperm throughout the stratification process. This migration was especially pronounced at the two key nodes of dormancy lifting. Mg is able to bind ATP for energy metabolism and stabilize ribosome attachment and activity to influence protein synthesis [58]. We suggest that the translocation of Mg in the seed is necessary to support the increased metabolic levels and protein

consumption of the seed embryo and endosperm during dormancy lifting. Ca can cross-link low-methyl esterified pectin in the cell wall, leading to the formation of a semi-rigid cell wall structure. Meanwhile, the reduction of Ca promotes the presence of highly methyl-esterified pectin. This, in turn, accelerates the degradation of the cell wall [59–61]. The relaxation of the cell walls of the seed embryo and endosperm weakens the resistance to seed germination. Data obtained from our laboratory (unpublished) showed that the activity of pectin methylesterase in the endosperm and seed embryo increased in the late stratification stage, and pectin was further degraded. Therefore, we believe that the Ca content in *P. hexandrum* embryos and endosperm decreased significantly in the late stratification stage. This promoted the degradation of cell wall pectin, destroyed the rigid structure of the cell wall, and accelerated the release of dormancy. Mn is the activator and cofactor of hundreds of metalloenzymes in plants and plays an important role in enzyme-catalyzed reactions, such as redox reactions, phosphorylation, decarboxylation, and hydrolysis. Simultaneously, Mn can activate SOD, therefore, balancing free radicals in plants [62, 63]. In the present study, Mn was mainly enriched in *P. hexandrum* seed embryos, indicating that the embryo is a core component of metabolic activity. The attack of intracellular oxygen free radicals leads to the relaxation of cell walls [64]. It is likely that after 45 d of stratification, a large amount of free radicals may be produced in the embryo to promote cell wall relaxation. The Mn element in the embryo may activate the synthesis of SOD to eliminate the toxic effects of excess ROS. In this study, we found that Co in seeds of *P. hexandrum* was mainly enriched in the seed embryo, and its content was higher when it was not stratified. Given that Co can block the conversion of ACC to ethylene by inhibiting the activity of ACC oxidase in the pathway of ethylene biosynthesis [65, 66], which is a prerequisite for the breaking of dormancy [67], a large amount of enriched Co in the seeds when they naturally matured could cause the acquisition of seed dormancy. This occurs through the inhibition of ethylene synthesis to be used to circumvent the unsuitable external environment after seed maturation. Meanwhile, the Co content in the seed embryos continued to decrease and that in the seed coat continued to increase as stratification proceeded. This migration of Co was an adaptive mechanism after seed sensing the environment. The decrease in Co content in seed embryo implied the increase in ethylene synthesis and the further lifting of dormancy. Our study on element accumulation and transport was determined by content differentiation. However, this method could not visually show the uptake and translocation of elements in the seeds. In the next step, we will use some visualization tools for further study to improve our understanding of this process.

In summary, we tentatively explored the link between dormancy release and changes in phytochemicials of *P. hexandrum* seeds, as well as the accumulation and migration of inorganic elements. Simultaneously, the mechanisms underlying the effects of these factors on seed dormancy release were tentatively speculated. Some key phytochemicals and inorganic elements were screened to provide references for the further development of new dormancy-breaking techniques. However, the actual dose–effect relationships need to be further verified experimentally.

## Acknowledgments

We thank Wenguang Zhang and Yanbin Xue for their help in writing and data processing, etc. Thanks to Gansu University of Chinese Medicine's undergraduate teaching practice center for providing the instruments and locations. We would like to thank Editage (www.editage.cn) for English language editing.

## Author Contributions

**Conceptualization:** Honggang Chen, Tao Du.

**Data curation:** Xijia Jiu, Honggang Chen, JinJin Meng.

**Formal analysis:** Xijia Jiu, XiWei Jia, Dong Liu, JinJin Meng.

**Funding acquisition:** Tao Du.

**Investigation:** XiWei Jia, XiaoJuan Xu.

**Methodology:** Honggang Chen, Tao Du.

**Project administration:** Honggang Chen, Tao Du.

**Resources:** Xijia Jiu, Dong Liu, XiaoJuan Xu.

**Software:** Xijia Jiu, XiWei Jia, Dong Liu.

**Supervision:** Tao Du.

**Writing – original draft:** Xijia Jiu.

**Writing – review & editing:** Xijia Jiu.

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
