## [Decision Letter · Decision Letter 0]

4 Sep 2023

PONE-D-23-09689Dormancy release from seeds of Sinopodophyllum hexandrum (Royle) Ying accompanied by changes in endogenous components and inorganic elements

PLOS ONE

Dear Dr. Du,

Thank you for submitting your manuscript to PLOS ONE. After careful consideration, we feel that it has merit but does not fully meet PLOS ONE’s publication criteria as it currently stands. Therefore, we invite you to submit a revised version of the manuscript that addresses the points raised during the review process.

We look forward to receiving your revised manuscript.

Kind regards,

Branislav T. Šiler, Ph.D.

Academic Editor

PLOS ONE

Journal Requirements:

" ext-link-type="uri" xlink:type="simple">https://journals.plos.org/plosone/s/file?id=ba62/PLOSOne_formatting_sample_title_authors_affiliations.pdf"

2.Thank you for stating the following in your Competing Interests section:  

"The authors declare that they have no known competing financial interests or personal relationships that could have appeared to influence the work reported in this paper."

3.Thank you for stating the following financial disclosure: 

"Supported by the earmarked fund for CARS-21."  

Additional Editor Comments:

Plant nomenclature must be in line with the scientifically accepted authorities such as Kew Garden's Plants of the World Online (https://powo.science.kew.org/) and World Flora Online (https://www.worldfloraonline.org/). Therefore, the model species *Sinopodophyllum hexandrum* (Royle) Ying is in fact just a synonym of the internationally accepted name *Podophyllum hexandrum* Royle (https://powo.science.kew.org/taxon/urn:lsid:ipni.org:names:77137821-1 and https://www.worldfloraonline.org/taxon/wfo-0000507972) and the latter should be used throughout the text, but the synonym can be mentioned in the Introduction section.

Several additional explanations are needed according to the reviewers' reports.

Figures should be prepared according to the PLOS ONE guidelines (https://journals.plos.org/plosone/s/figures) and submitted as separate files in the revised version.

The Discussion section must be improved to contain active and thorough comparison of the results obtained in the study and those find in the literature.

Language usage must be substantially improved since in some parts the text is incomprehensible; the manuscript needs to be proofread by a native English speaker or a professional editing agency.

Reviewers' comments:

Reviewer's Responses to Questions

**Comments to the Author**

1. Is the manuscript technically sound, and do the data support the conclusions?

Reviewer #1: Partly

Reviewer #2: Yes

2. Has the statistical analysis been performed appropriately and rigorously? 

Reviewer #1: Yes

Reviewer #2: Yes

3. Have the authors made all data underlying the findings in their manuscript fully available?

Reviewer #1: Yes

Reviewer #2: Yes

4. Is the manuscript presented in an intelligible fashion and written in standard English?

Reviewer #1: No

Reviewer #2: No

5. Review Comments to the Author

Reviewer #1: In the present manuscript, the authors investigated the changes in the contents of inorganic elements and biomolecules in endosperm and seed coat during dormancy release in Sinopodophyllum hexandrum. A lot of effort emphasized proteins and transcriptional changes during seed dormancy release in this plant. However, the dynamics of other biomolecules including lipids and inorganic elements is almost unexplored. As inorganic elements facilitate proteins and enzyme in their functioning and other biomolecules also regulate seed germination process, the concenption of the manuscript have merits.

Authors analaysed the changes in endogenous components and inorganic elements in different seed parts during cold stratification mediated dormancy release, and tried to implicate those changes with dormancy release/seed germination.

Though the comnception has merits, the manuscript needs thorough improvements.

Please see comments below:

Comments:

• Writing needs a thorough editing and improvemt throughout.

• “Endogenous components” doesn’t specifies any thing and looks vague. This shoud be replaced with more suitable words such as “phytochemicials” or “phytomolecules”.

• Discussion is very poor. There’s no inference collected and simply previous literature is cited.

• It looks that authors just presented the profilings data, and drawn no inferences.

• There’s no explanation how various phytomolcules and inorganic elemnst present in seed coat, endosperm and embryo, could be functioning in dormancy release or seed germination.

Reviewer #2: Sinopodophyllum hexandrum is a medicinal plant that produces podophyllotoxin, a compound with anti-cancer properties. However, the plant is endangered due to over-harvesting and habitat loss. Therefore, seed germination and propagation studies are important for its conservation and sustainable use. The MS findings lies within the scope of the journal PLOS ONE. The MS aimed at understanding the factors that affect the seed dormancy release and seed germination of S. hexandrum. Though it is well known that seed dormancy in S. hexandrum is mainly caused by the presence of a hard seed coat that prevents water uptake and gas exchange, seed dormancy can be broken by cold treatment. However, its physiological bases are poorly understood. Some studies showed that S. hexandum showed dormancy due to under developed embryo as well as physiological factors such as ABA/GA, antioxidants, H2O2 that help in maintaining GA/ABA ratio. Here I have some questions.

1. If seed dormancy in S. hexandrum is due to both factors (under developed embryo, and biochemicals hormones), the authors applied only cold treatment. Literature showed that cold treatment 4 oC for 60-90 days and 30-60 days treatment of 25 oC is required. Why the authors do not follow the standard protocol for treatment and analyze the biochemicals over 15 days interval.

2. Why authors do not consider the analysis of hormones such as ABA, GA and expression of genes responsible for biosynthesis/catabolism of these hormones

My general observations are as follows:

Introduction: It is suggested to add review literature about biochemical/physiological factors in seed dormancy and release and cite the reports about which metabolites may affect GA/ABA ratio etc.

Results: English of this section is poor and it seems that senior author did not rectify this section. For example, very difficult to understand these sentences

As shown in Figure 1A, the GP of S.hexandrum seeds grew from the initial 34.81% to 87.22%

217 with the lengthening of stratification time, and the dormancy was gradually released. As

218 illustrated in Figure 1B, the T50 of seeds at each stage presented an overall downward trend,

219 but the variation range was small. After being stratified for 90 d, the T50 of seeds was

220 significantly lower than that at other stages.

Discussion It is suggested to add one paragrph about % release of seed dormancy with days after cold treatment, which will be followed by discussion on migration of nutrients and metabolites from seed part and release of seed dormancy.

Overall, the MS is well written and can be accepted after minor changes in the MS.

6. PLOS authors have the option to publish the peer review history of their article (what does this mean?). If published, this will include your full peer review and any attached files.

Reviewer #1: No

Reviewer #2: **Yes: **Prof Dr Habib-ur-Rehman Athar

---

## [Author Response · Author response to Decision Letter 0]

16 Oct 2023

We thank the editors and reviewers for their guidance and helpful comments. According to suggestions, we have made a lot of changes to the full text, Includes language editing, reopening the discussion, modifying inappropriate descriptions, rethinking and summarizing results, and point-by-point responses to each comment.

Editor Comments:

1.Plant nomenclature must be in line with the scientifically accepted authorities such as Kew Garden's Plants of the World Online (https://powo.science.kew.org/) and World Flora Online (https://www.worldfloraonline.org/). Therefore, the model species Sinopodophyllum hexandrum (Royle) Ying is in fact just a synonym of the internationally accepted name Podophyllum hexandrum Royle (https://powo.science.kew.org/taxon/urn:lsid:ipni.org:names:77137821-1 and https://www.worldfloraonline.org/taxon/wfo-0000507972) and the latter should be used throughout the text, but the synonym can be mentioned in the Introduction section.

Authors’ Response: We have completely revised the Latin names in the manuscript

2.Figures should be prepared according to the PLOS ONE guidelines (https://journals.plos.org/plosone/s/figures) and submitted as separate files in the revised version.

Authors’ Response: I have reorganized the figures in the manuscript according to the style of the journal.

3.The Discussion section must be improved to contain active and thorough comparison of the results obtained in the study and those find in the literature.

Authors’ Response:We followed the comments and revised and refined 80% of the discussion, introducing more positive literature to try to elaborate on some of the positive associations between the results and give some preliminary inferences about their underlying mechanisms. The revised discussion section will be shown in response to Reviewer #1.

4.Language usage must be substantially improved since in some parts the text is incomprehensible; the manuscript needs to be proofread by a native English speaker or a professional editing agency.

Authors’ Response: We took the journal's advice and used Editage's touch-up service. The resubmitted manuscript has been revised and improved in its entirety by a native-speaking editor with a PhD in the field, However, due to the excessive number of minor changes involving language throughout the text, I have only highlighted the more altered language issues in the manuscript.

Reviewer #1:

1.Writing needs a thorough editing and improvemt throughout.

Authors’ Response: We submitted the manuscript to Editage and used professional touch-up services, and the resubmitted manuscript was revised and refined by a native-speaking editor with a Ph.D. in the field, in addition to inviting other experts in the field from our institution to further review the manuscript.

2.Endogenous components” doesn’t specifies any thing and looks vague. This shoud be replaced with more suitable words such as “phytochemicials” or “phytomolecules”.

Authors’ Response: Thank you for your valuable comments, which we have taken on board and replaced the term endogenous components in the manuscript with phytochemicials.

3.Discussion is very poor. There’s no inference collected and simply previous literature is cited.

4.It looks that authors just presented the profilings data, and drawn no inferences.

5.There’s no explanation how various phytomolcules and inorganic elemnst present in seed coat, endosperm and embryo, could be functioning in dormancy release or seed germination.

Authors’ Response: Thank you for these three valuable points! We believe that these three comments have a similar kernel, so we have put them together in order to answer them. We have reconsidered and reorganized the relationships between the data, made extensive changes to the original discussion, including the structure of the presentation, the linkage between the data and the existing literature, etc., and extrapolated and dissected the possible mechanisms of how phytochemicals and inorganic elements work to undo dormancy through a number of positive literatures, and since the discussion portion of the discussion has undergone almost complete changes, we are attaching the revised discussion here to make it easier for you to review it further.

Revised discussion: Low-temperature stratification is a well-established and efficient method for breaking seed dormancy, particularly physiological dormancy. We assessed the dormancy levels of seeds at various stratification durations using four indices, that is, the germination rate, germination time lag, water absorption rate, and water absorption rate. We discovered that 90 d of low-temperature stratification effectively lifted the dormancy of P. hexandrum seeds. The entire stratification process could be divided into three levels based on the degree of dormancy. The seed germination rate could reach 87.22% after 90 d of stratification, and with the gradual lifting of dormancy, the time lag of germination was shortened and the rate of water uptake was accelerated. This indicated that the permeability of the seed coat increased, and the improved germination ability was also an indicator of an increase in the level of physiological metabolism within the seeds. 

Phytochemicals in plants affect seed germination; terpenes, ketones, phenols, and fatty acids are common examples. Previous studies have provided direct evidence of this. Eight sesquiterpenes in the Acorus calamus seeds inhibit the germination of lettuce seeds [36]. The ketones (1S, 6R)-2, 7 (14), and 10-bisaboltrien-1-ol-4-one in Cryptomeria japonica seeds display inhibitory activity against lettuce and rice seeds [37]. α-pyrone derivatives can completely inhibit the germination of Amaranthus caudatus seeds [38]. Catechin, a phenolic component in Lespedeza seeds, represses seed germination [39]. Seed dormancy is also influenced by phytochemicals in the seeds. Dormancy of lettuce seeds may be induced by the accumulation of coumarin [40]. Meanwhile, dormancy of Arabidopsis thaliana seeds is promoted by 12-oxo-plant dienoic acid and abscisic acid [41]. The composition of phytochemicals in the seeds varies considerably among different plants. Phenols and esters are mainly contained in the seed coat of Triticum aestivum [42]. Meanwhile, phenolic acids, such as p-hydroxybenzoic acid and protocatechuic acid, play a dominant role in the seed coat of Parthenium argentatum [43]. According to the GC-MS analysis of methanol extracts from the seed coat and endosperm at different stratification stages, the phytochemicals were mainly alkanes and fatty acids, the types and contents of which varied with different seed parts. More phenols were detected in the seed coat, whereas more ketones and esters were present in the endosperm. This may have had different effects on dormancy release. Following multivariate statistical analysis, five key differential components were filtered from the seed coat and endosperm and were found to be ketones, esters, alkanes, and fatty acids. A variety of derivatives with pyranone as the parent nucleus exhibit strong free radical scavenging and antioxidant functions [44].Separation of fractions and evaluation of antioxidant activity of leaf extracts of Odontonema strictum showed that 6-substituted 5,6-dihydro-α-pyrones were the major components exerting antioxidant activity [45]. The activities of superoxide dismutase, peroxidase, and catalase were significantly higher in the exogenous pyrantel treatment group than the control [46]. reactive oxygen metabolism play an important role in the regulation of seed dormancy. The biosynthesis of GA is stimulated by ROS through mitogen-activated protein kinase (MAPK) cascades. The accumulation of H2O2 causes ABA degradation through influencing ABA catalytic enzyme [47]. Therefore it is important to maintain the homeostasis of reactive oxygen species in seeds. In this study, we found that the ketone components 2,3-Dihydro-3,5-dihydroxy-6-methyl-4(H)-pyran-4-one and 6-methyl-3-(1-methylethyl)-7-oxabicyclo[4.1.0]heptan-2-one in the endosperm of P. hexandrum seed showed an increasing trend at the late stage of stratification. oxabicyclo[4.1.0]heptan-2-one both showed an increasing trend in the late stage of stratification. This trend was significantly positively correlated with their seed germination rates and significantly negatively correlated with T50. From this, we deduced that the seeds of P. hexandrum showed an increasing trend in the late stage of stratification through the accumulation of ketone constituents, 2,3-Dihydro-3,5-dihydroxy-6-methyl-4(H)-pyran-4-one and 6-methyl-4(H)-pyran-4-one. (H)-pyran-4-one and 6-methyl-3-(1-methylethyl)-7-oxabicyclo[4.1.0]heptan-2-one, therefore, enhancing their own antioxidant capacity, eliminating excessive reactive oxygen species during metabolism, maintaining reactive oxygen species homeostasis, and accelerating the rate of seed germination and facilitating the lifting of dormancy. Recent experimental data (unpublished) also showed that SOD activity and CAT activity within the endosperm of P. hexandrum seeds would rise significantly in the late stage of stratification, which supports this inference. Alkanes are the main components of cuticle wax [48,49]. As a hydrophobic layer, wax usually prevents water uptake, which affects germination. In this study, we found that the contents of alkane components 6-Propyltridecane and Hexadecane in the seed coat of P. hexandrum showed a decreasing trend during lamination and were positively correlated with the T50 of P. hexandrum seeds. We hypothesized that the decrease in the contents of 6-Propyltridecane and Hexadecane resulted in the loosening of the cuticle waxes and enhanced the hydrophilicity of the seed coat. This, in turn, increased seed permeability and promoted germination. Alkane synthesis is regulated by phytohormones and transcription factors, and ABA and Melatonin act synergistically to promote alkane synthesis in watermelon leaves under drought stress [50]. The mechanism involved may be ABA promoting alkane biosynthesis by negatively regulating the transcriptional repressors of BnCER1-2, BnaC9.DEWAX1. This , in turn, promotes alkane biosynthesis [51 Changes in phytohormones within the seeds of P. hexandrum during low-temperature stratification [13] had an overall decreasing trend in the ABA content. This is also in line with the results that we have obtained thus far. However, further experiments are needed to verify whether ABA mediates the synthesis of alkane constituents, such as 6-Propyltridecane and Hexadecane. This, in turn, modulates the translucency of the seed coat of P.hexandrum. Fatty acids are thought to act as chemosensitizers that affect seed germination. Short-chain fatty acids inhibit the germination of Cucumis sativus seeds [52] and maintain the dormancy of Avena fatua seeds [53]. Meanwhile, the premature germination of Mangifera indica seeds results from a reduction in long-chain fatty acid synthesis [54]. The exogenous application of palmitic and linoleic acid monomers significantly inhibited the germination of Lolium multiflorum and Phalaris minor seeds, and the inhibitory ability increased with increasing concentration [55].In this study, the inhibitory effects of the seed coat extract and endosperm extract of peach seed on the germination of oilseed rape seeds at different stages of stratification were consistent with the changes in contents of palmitic acid and linoleic acid in their corresponding parts. The contents of palmitic acid and linoleic acid accounted for a larger proportion in both the seed coat and the endosperm. Therefore, it is likely that these two phytochemicals are the main germination inhibitors in P. hexandrum seed. Saccharides are the end products of fatty acid decomposition. Through a series of pathways such as oxidation, fatty acids, the glyoxylate cycle, and the tricarboxylic acid cycle, ultimately generate glucose and sucrose by gluconeogenesis. This can be used as a carbon source and an energy donor to directly promote seed germination of seeds [56]. We hypothesized that the reduced linoleic and palmitic acid content in the seed coat and endosperm of P. hexandrum seeds is broken down into soluble sugars for direct energy supply. This leads to a significant increase in seed germination in the later stages of stratification.

The release of seed dormancy is concurrent with intense physiological activities and energy metabolism, such as the decomposition of substances, such as starch, lipids, and proteins, during storage, the activation of related metabolic enzymes, and the synthesis of physiologically active substances [57], with which the physiological functions of inorganic elements are often closely associated. We first determined the changes in the contents of 11 elements, including Mg, Ca, Mn, Zn, and Cu, which are commonly thought to be beneficial elements, in P. hexandrum seed coats at various stages of stratification. We then screened out the five differential inorganic elements, that is, Ca, Mg, Mn, Co, and Se, by combining the PLS-DA analysis with the T-test. This suggested that these five elements may play an important role in the process of dormancy lifting. Correlation research also showed that there may be a synergistic effect between Mg, Mn, and Co. However, it is still unclear how inorganic elements move and accumulate in different sections of the seed. Therefore, we determined the content of the five main differential elements in stages and parts. The Mg element in seeds was primarily enriched in the seed coat prior to stratification. However, the Mg content in the seed embryo and endosperm increased significantly in the late stage of stratification. This suggests that there was a trajectory of Mg migration from the outer seed coat to the inner seed embryo and endosperm throughout the stratification process. This migration was especially pronounced at the two key nodes of dormancy lifting. Mg is able to bind ATP for energy metabolism and stabilize ribosome attachment and activity to influence protein synthesis [58]. We suggest that the translocation of Mg in the seed is necessary to support the increased metabolic levels and protein consumption of the seed embryo and endosperm during dormancy lifting. Ca can cross-link low-methyl esterified pectin in the cell wall, leading to the formation of a semi-rigid cell wall structure. Meanwhile, the reduction of Ca promotes the presence of highly methyl-esterified pectin. This, in turn, accelerates the degradation of the cell wall [59-61]. The relaxation of the cell walls of the seed embryo and endosperm weakens the resistance to seed germination. Data obtained from our laboratory (unpublished) showed that the activity of pectin methylesterase in the endosperm and seed embryo increased in the late stratification stage, and pectin was further degraded. Therefore, we believe that the Ca content in P. P. hexandrum embryos and endosperm decreased significantly in the late stratification stage. This promoted the degradation of cell wall pectin, destroyed the rigid structure of the cell wall, and accelerated the release of dormancy. Mn is the activator and cofactor of hundreds of metalloenzymes in plants and plays an important role in enzyme-catalyzed reactions, such as redox reactions, phosphorylation, decarboxylation, and hydrolysis. Simultaneously, Mn can activate SOD, therefore, balancing free radicals in plants [62,63]. In the present study, Mn was mainly enriched in P. hexandrum seed embryos, indicating that the embryo is a core component of metabolic activity. The attack of intracellular oxygen free radicals leads to the relaxation of cell walls [64]. It is likely that after 45 d of stratification, a large amount of free radicals may be produced in the embryo to promote cell wall relaxation. The Mn element in the embryo may activate the synthesis of SOD to eliminate the toxic effects of excess ROS. In this study, we found that Co in seeds of P. hexandrum was mainly enriched in the seed embryo, and its content was higher when it was not stratified. Given that Co can block the conversion of ACC to ethylene by inhibiting the activity of ACC oxidase in the pathway of ethylene biosynthesis [65,66], which is a prerequisite for the breaking of dormancy [67], a large amount of enriched Co in the seeds when they naturally matured could cause the acquisition of seed dormancy. This occurs through the inhibition of ethylene synthesis to be used to circumvent the unsuitable external environment after seed maturation. Meanwhile, the Co content in the seed embryos continued to decrease and that in the seed coat continued to increase as stratification proceeded. This migration of Co was an adaptive mechanism after seed sensing the environment. The decrease in Co content in seed embryo implied the increase in ethylene synthesis and the further lifting of dormancy. Our study on element accumulation and transport was determined by content differentiation. However, this method could not visually show the uptake and translocation of elements in the seeds. In the next step, we will use some visualization tools for further study to improve our understanding of this process.

In summary, we tentatively explored the link between dormancy release and changes in phytochemicials of P. hexandrum seeds, as well as the accumulation and migration of inorganic elements. Simultaneously, the mechanisms underlying the effects of these factors on seed dormancy release were tentatively speculated. Some key phytochemicals and inorganic elements were screened to provide references for the further development of new dormancy-breaking techniques. However, the actual dose–effect relationships need to be further verified experimentally.

Reviewer #2:

1. If seed dormancy in S. hexandrum is due to both factors (under developed embryo, and biochemicals hormones), the authors applied only cold treatment. Literature showed that cold treatment 4 ℃ for 60-90 days and 30-60 days treatment of 25 ℃ is required. Why the authors do not follow the standard protocol for treatment and analyze the biochemicals over 15 days interval.

Authors’ Response: The seeds of S. hexandrum have different types of dormancy in different ecological environments, and we have investigated the causes of dormancy in S. hexandrum in our previous experiments, and examined the morphology of seed embryo, the changes of phytohormone content, and the activities of key enzymes in the tricarboxylic acid cycle in S. hexandrum, etc. As far as our collection site is concerned, the seeds of S. hexandrum from Hezheng Medicinal Botanical Garden have been found to be in the following types of dormancy in our experiments for two consecutive years Physiological dormancy, and there is no morphological dormancy, the seeds were dissected after maturity and found that the seed embryo was mature embryo, and the relevant data have been published (Changes of Embryo Morphology, Physiology and Biochemistry during Low Temperature Stratification of S. hexandrum Seeds. Acta Botanica Boreali-Occidentalia Sinica. 2021;12: 2096.), it is well known that warm stratification promotes the development of seed embryo, but there was no morphological dormancy in our material, so we only performed low temperature stratification. In addition, we also compared the ability of different stratification procedures to release dormancy in the preliminary stage, and found that warm stratification had little effect on releasing its dormancy, while low-temperature stratification at 4°C for 90 d was the most effective method, so we chose this method in this study, and sampling at 15-d intervals was decided based on the results of the preliminary test and the prominent changes in germination rate.

2. Why authors do not consider the analysis of hormones such as ABA, GA and expression of genes responsible for biosynthesis/catabolism of these hormones

Authors’ Response: In our previous study, we have analyzed the changes of phytohormones within the seeds of S. hexandrum during dormancy lifting, including GA, IAA, ABA, and so on, and also analyzed the proportionality among these hormones, and we found that with dormancy lifting, its IAA content increased, ABA content decreased, and the proportions of GA/ABA, IAA/ABA, and GA + IAA/ABA increased, which was in line with other researchers, therefore, we did not focus on the hormone changes in this study. In addition, there is an exciting news to share with you, during our field work, we found a natural low-dormancy S. hexandrum germplasm resource, and we are going to compare the differences in transcriptome and metabolome between the original germplasm and the newly discovered germplasm in the next step of our work, which may be useful for the research. This may be of great significance to our discovery of the core components that regulate seed dormancy in S. hexandrum.

3.Introduction: It is suggested to add review literature about biochemical/physiological factors in seed dormancy and release and cite the reports about which metabolites may affect GA/ABA ratio etc.

Authors’ Response: Thank you for your valuable suggestions, I have followed the comments and added a paragraph to the introduction, the details of which are attached here for your easy review.

New additions:The end of seed dormancy depends on changes in the biochemical levels. Various phytochemicals within the seed, including lipids, starches, proteins, amino acids, and phytohormones, migrate, accumulate, and transform during this process. Amino acids such as arginine serve as nitrogen sources for protein synthesis. Lipids, such as triglycerides, are converted into energy-supporting substances through participation in either the tricarboxylic acid cycle (TCA) or pyruvic acid pathway. Both of these pathways accumulate in large quantities during germination, promoting breaking dormancy [19]. Phytohormones play a crucial role in the regulation of seed dormancy. Gibberellins (GA) and Abscisic acid (ABA) are antagonistic. GA promotes seed dormancy by stimulating the synthesis of Aux and Cytokinin (CTK) as well as by activating amylase and protease. In contrast, ABA induces dormancy by regulating the accumulation of storage proteins and lipids in the seed [20]. Phytohormones migrate between different tissue parts of the seed. ABA is produced in the endosperm and transported to the seed embryo, whereas GA is produced in the seed embryo and transported to the endosperm during seed germination. GA activates carbon metabolism in the endosperm and facilitates the translation and synthesis of critical proteins for cell growth [21]. The synthesis of phytohormones is usually associated with a number of other metabolites or metabolic pathways. ABA is produced from the key precursor zeaxanthin via a series of epoxidation and isomerisation reactions. GA is produced via a dioxygenase reaction with the key intermediate 2-oxyglutarate in the TCA pathway. In the phytoterpene synthesis pathway, the precursors isopentenyl diphosphate (IPP) and dimethylallyl diphosphate (DMAPP), and the precursors in the plant terpene synthesis pathway, are the basis for the synthesis of these two key phytohormones. Therefore, the synthesis of these two hormones competes with the synthesis of terpenes. Jasmonate (JA) synthesis begins with the formation of 12-oxo-phytodienoic acid from the oxidation of linolenic acid. The synthesis and content of linolenic acid, to a certain extent, determine the level of JA in plants [22]. Many transcriptomic and metabolomic studies have provided evidence for the relationship between phytochemicals and seed dormancy. Studies on Heracleum moellendorffii Hance seeds have shown that high expression of enoyl-CoA hydratase promotes the accumulation of fatty acids in the seeds. This facilitates the development of the embryo, and, therefore, accelerates the release of dormancy [23]. Comparative metabolomic studies of wheat seeds at two levels of dormancy have shown that unsaturated fatty acid analogs, such as cis-vaccenate, oleate, linoleate, and linolenate, undergo accumulation mediated by phospholipase A2 to maintain dormancy. Meanwhile, oxalate regulates the accumulation of fatty acids in seeds, likely through its involvement in ROS metabolism. It is involved in ROS metabolism to regulate seed dormancy, and higher oxalate levels imply an increase in the level of lipid degradation, which promotes the lifting of dormancy [24].

4. Results: English of this section is poor and it seems that senior author did not rectify this section. For example, very difficult to understand these sentences

As shown in Figure 1A, the GP of S.hexandrum seeds grew from the initial 34.81% to 87.22%

217 with the lengthening of stratification time, and the dormancy was gradually released. As

218 illustrated in Figure 1B, the T50 of seeds at each stage presented an overall downward trend,

219 but the variation range was small. After being stratified for 90 d, the T50 of seeds was

220 significantly lower than that at other stages.

Authors’ Response: We were deeply concerned that language was one of the big problems with this manuscript, so we used the touch-up services of Editage, a professional touch-up team, with the full text revised and refined by experts in the field with Ph.D.'s and native speakers, and we invited senior scholars in our organization to review and refine the manuscript.

5. Discussion It is suggested to add one paragrph about % release of seed dormancy with days after cold treatment, which will be followed by discussion on migration of nutrients and metabolites from seed part and release of seed dormancy.

Authors’ Response: Thank you for your valuable suggestions, I have followed the comments and added a paragraph to the discussion, the details of which are attached here for your easy review.

New additions: Low-temperature stratification is a well-established and efficient method for breaking seed dormancy, particularly physiological dormancy. We assessed the dormancy levels of seeds at various stratification durations using four indices, that is, the germination rate, germination time lag, water absorption rate, and water absorption rate. We discovered that 90 d of low-temperature stratification effectively lifted the dormancy of P. hexandrum seeds. The entire stratification process could be divided into three levels based on the degree of dormancy. The seed germination rate could reach 87.22% after 90 d of stratification, and with the gradual lifting of dormancy, the time lag of germination was shortened and the rate of water uptake was accelerated. This indicated that the permeability of the seed coat increased, and the improved germination ability was also an indicator of an increase in the level of physiological metabolism within the seeds.

---

## [Editor Report · Decision Letter 1]

27 Oct 2023

PONE-D-23-09689R1Dormancy release of seeds of Podophyllum hexandrum Royle accompanied by changes in phytochemicals and inorganic elementsPLOS ONE

Dear Dr. Du,

Thank you for submitting your manuscript to PLOS ONE. After careful consideration, we feel that it has merit but does not fully meet PLOS ONE’s publication criteria as it currently stands. Therefore, we invite you to submit a revised version of the manuscript that addresses the points raised during the review process.

Although considerably improved according to the reviewers' reports, the manuscript still cannot be accepted due to several minor issues:

L35, 36-37 and elsewhere in the text: Once fully introduced, the plant species' name should be shortened to* P. hexandrum* containing no authority name. Full species name can remain in its first mention in Abstract, Introduction and MM. The same applies to L111.

L39-40: Please choose keywords not mentioned in the main title that are closely associated to the study theme.

L42: "*Podophyllum hexandrum* is a perennial *Sinopodophyllum* Ying herb that is..." cannot stand, since a species cannot belong to two genera. Please rewrite as suggested in the first review round.

L50: *Podophyllum hexandrum* is not listed in the IUCN Red List under any threatened category. Please remove this remark.

L77,78, and elsewhere in the text: Please do not capitalize compound names or randomly chosen words (see the attached file where I highlighted SOME BUT NOT ALL places of intervention).

L130-131: Please provide geographic coordinated in the internationally accepted format.

L177, 179: some strange signs stand here.

L285: What "The same as below" means?

Fig 4, Fig 5, Fig 6, Fig 7, Fig 9, Fig 10, Fig 11: Provide here informative captions. They should contain short descriptions of what can be seen from the bars, dots, etc. and description of what bars/dots were compared and their statistical significance.

Fig 8: Please provide better descriptions of what can be seen from the graphs and provide PC components' contribution.

We look forward to receiving your revised manuscript.

Kind regards,

Branislav T. Šiler, Ph.D.

Academic Editor

PLOS ONE
---

## [Author Response · Author response to Decision Letter 1]

4 Nov 2023

We thank the academic editors for guidance and helpful comments. According to suggestions, we have made a lot of changes to the full text, includes abbreviation of species' name, case-sensitivity of text, standardization of geographic information, correction of clerical errors throughout the text and better description of figures.

Academic editor Comments:

1. L35, 36-37 and elsewhere in the text: Once fully introduced, the plant species' name should be shortened to P. hexandrum containing no authority name. Full species name can remain in its first mention in Abstract, Introduction and MM. The same applies to L111.

Authors’ Response: We have completely revised the species' name in the manuscript.

2. L39-40: Please choose keywords not mentioned in the main title that are closely associated to the study theme.

Authors’ Response: We replaced the keywords as per the comments.

3. "Podophyllum hexandrum is a perennial Sinopodophyllum Ying herb that is..." cannot stand, since a species cannot belong to two genera. Please rewrite as suggested in the first review round..

Authors’ Response: Thanks for the detailed suggestions, we have made careful corrections and modifications.

4. Podophyllum hexandrum is not listed in the IUCN Red List under any threatened category. Please remove this remark.

Authors’ Response: Thanks for the valuable suggestion, we have removed this description.

5.L77,78, and elsewhere in the text: Please do not capitalize compound names or randomly chosen words (see the attached file where I highlighted SOME BUT NOT ALL places of intervention).

Authors’ Response: Thank you very much for your suggestions, we have made very careful changes!

6. L130-131: Please provide geographic coordinated in the internationally accepted format.

Authors’ Response: Thank you for your suggestions, we have made changes to the geographic information.

7.L177, 179: some strange signs stand here.

Authors’ Response: Thank you very much for your careful review, we have been carefully revised.

8.L285: What "The same as below" means?

Authors’ Response: Thank you very much for your careful review, we have removed this description.

9.Fig 4, Fig 5, Fig 6, Fig 7, Fig 9, Fig 10, Fig 11: Provide here informative captions. They should contain short descriptions of what can be seen from the bars, dots, etc. and description of what bars/dots were compared and their statistical significance.

Authors’ Response: Thanks to your valuable suggestions, we have added short informative notes and additional explanations for some of the information in the diagrams.

10.Fig 8: Please provide better descriptions of what can be seen from the graphs and provide PC components' contribution.

Authors’ Response: Thanks to your valuable suggestions, in order to better describe the content of the graphs, we have added explanations of the data in the graphs and added their statistical significance.

---

## [Editor Report · Decision Letter 2]

7 Nov 2023

Dormancy release of seeds of Podophyllum hexandrum Royle accompanied by changes in phytochemicals and inorganic elements

PONE-D-23-09689R2

Dear Dr. Du,

We’re pleased to inform you that your manuscript has been judged scientifically suitable for publication and will be formally accepted for publication once it meets all outstanding technical requirements.

Kind regards,

Branislav T. Šiler, Ph.D.

Academic Editor

PLOS ONE
---

## [Editor Report · Acceptance letter]

9 Nov 2023

PONE-D-23-09689R2 

Dormancy release of seeds of *Podophyllum hexandrum* Royle accompanied by changes in phytochemicals and inorganic elements 

Dear Dr. Du:

I'm pleased to inform you that your manuscript has been deemed suitable for publication in PLOS ONE. Congratulations! Your manuscript is now with our production department. 

Kind regards, 

on behalf of

Dr. Branislav T. Šiler 

Academic Editor

PLOS ONE